# Direct tracking of H₂ roaming reaction in real time

**Debadarshini Mishra** [1,6] ✉, **Aaron C. LaForge** [1,6] ✉, **Lauren M. Gorman**[1], **Sergio Díaz-Tendero** [2,3,4], **Fernando Martín** [2,3,5] **& Nora Berrah** [1]

Roaming is an unconventional type of molecular reaction where fragments, instead of immediately dissociating, remain weakly bound due to long-range Coulombic interactions. Due to its prevalence and ability to form new molecular compounds, roaming is fundamental to photochemical reactions in small molecules. However, the neutral character of the roaming fragment and its indeterminate trajectory make it difficult to identify experimentally. Here, we introduce an approach to image roaming, utilizing intense, femtosecond IR radiation combined with Coulomb explosion imaging to directly reconstruct the momentum vector of the neutral roaming $H_2$, a precursor to $H_3^+$ formation, in acetonitrile, $CH_3CN$. This technique provides a kinematically complete picture of the underlying molecular dynamics and yields an unambiguous experimental signature of roaming. We corroborate these findings with quantum chemistry calculations, resolving this unique dissociative process.

Understanding the multitude of complex molecular processes proceeding through photoexcitation is of fundamental importance to a wide variety of biological and chemical systems. One of the most common photochemical processes is dissociation, which typically proceeds by the fragmenting molecule moving along the minimum energy path (MEP). However, there exists a more complex pathway, known as roaming, where the neutral fragment spans the relatively flat regions of the potential energy surface, farther from the MEP, due to long-range, weakly-bound interactions. Roaming has been a subject of great interest in recent years[1-3] since its initial indirect observation in formaldehyde[4] and has been proposed to occur in several small molecules such as acetaldehyde[5,6], acetone[7], nitrate[8], methyl formate[9,10], propane[11], and 2-propanol[12].

Roaming is known to play a crucial role in the formation of $H_3^+$, one of the most abundant molecular ions in the universe[13,14]. $H_3^+$ formation dynamics is of great importance in gas-phase astrochemistry, due to its critical role as an intermediate in the synthesis of more complex molecules[15-17]. Very recent works[16,17] have specifically examined the formation of $D_3^+$, an isotope of $H_3^+$, from a bimolecular reaction in $D_2–D_2$ dimers as a potential explanation for its unexpectedly high abundance in interstellar molecular clouds. High-powered, ultrafast lasers have emerged as an efficient means of producing $H_3^+$ ions through photoexcitation of organic molecules and have thus facilitated further exploration of their dynamics[18-24]. Notably, roaming of $H_2$, triggered by intense femtosecond IR laser pulses, has been shown to result in $H_3^+$ formation in monohydric alcohols[25,26]. Complementary experiments, using XUV femtosecond laser pulses and Coulomb explosion imaging (CEI)[27-31], have also explored the role of this mechanism leading to $H_3^+$ formation. In both cases, roaming is initiated in the molecular dicationic state. IR pulses rely on strong-field ionization due to a strong coupling between the molecular system and the electric field of the laser, while XUV pulses can prompt single-photon double ionization. As such, roaming is quite ubiquitous and recently has also been suggested to occur in neutral formaldehyde, rather than from the dicationic state[32].

Despite recent efforts, the real-time, direct visualization of roaming has remained elusive from experimental observations due to its complex nature. The central problem is the random nature of the

[1]Department of Physics, University of Connecticut, Storrs, CT 06269, USA. [2]Departamento de Química, Módulo 13, Universidad Autónoma de Madrid, 28049 Madrid, Spain. [3]Condensed Matter Physics Center (IFIMAC), Universidad Autónoma de Madrid, 28049 Madrid, Spain. [4]Institute for Advanced Research in Chemical Sciences (IAdChem), Universidad Autónoma de Madrid, 28049 Madrid, Spain. [5]Instituto Madrileño de Estudios Avanzados en Nanociencia (IMDEA-Nano), Campus de Cantoblanco, 28049 Madrid, Spain. [6]These authors contributed equally: Debadarshini Mishra, Aaron C. LaForge. ✉e-mail: debadarshini.mishra@uconn.edu; aaron.laforge@uconn.edu

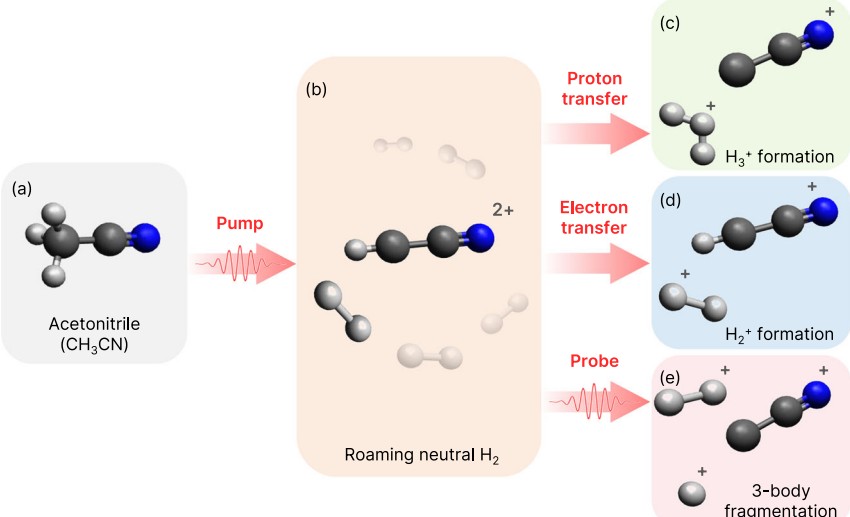

**Fig. 1 | Schematic representation of H$_2$ roaming induced by the pump pulse and the resultant product channels in CH$_3$CN (N atom in blue, C atoms in dark gray, and H atoms in light gray).** The pump pulse on (**a**) a gaseous sample of CH$_3$CN creates the parent dicationic state, [CH$_3$CN]$^{2+}$ through multiphoton absorption, followed by (**b**) neutral dissociation of H$_2$. A small subset of these events may lead to either (**c**) proton transfer (H$_3^+$ + C$_2$N$^+$) or (**d**) electron transfer (H$_2^+$ + C$_2$NH$^+$) without the involvement of the probe pulse. Conversely, the roaming H$_2$ may be ionized by the probe pulse, resulting in the formation of (**e**) H$^+$ + H$_2^+$ + C$_2$N$^+$ channel.

roaming process, where the position of the roaming particle relative to the rest of the system is not deterministic. For instance, pioneering experimental studies inferred the role of roaming neutral H$_2$ in H$_3^+$ formation by using time-resolved mass spectra[25,26]. In the absence of coincident momentum imaging, it is not possible to directly identify which fragmentation channels contributed to the formation of H$_3^+$. More recently, Endo et al.[32] demonstrated the time-resolved signature of roaming in neutral formaldehyde by investigating a three-ion coincidence channel using CEI. However, this channel carried imprints of various reactions that are triggered via UV excitation of formaldehyde, including molecular and radical dissociation, as well as roaming. As a result, it was necessary to incorporate high-level simulations to identify the signature of roaming from the experimental results.

In this work, we directly track the roaming process by experimentally measuring the complex dynamics of the neutral H$_2$, a crucial intermediate in the formation of H$_3^+$, in acetonitrile, CH$_3$CN. In general, the neutral character of the roaming fragment makes it invisible to charged-particle detection schemes, and further, its random trajectory makes it extremely difficult to track both spatially and temporally. However, with the use of coincident momentum imaging in combination with femtosecond 800 nm IR-IR pump-probe spectroscopy, we have access to the full, time-resolved 3D momentum information of each detected fragment. This enables us to reconstruct the momentum vector of the undetected neutral fragment in a reaction, by virtue of momentum conservation. Such kinematically complete information, even for a coincidence channel involving a neutral fragment, is crucial for mapping out a direct signature of roaming reactions without any interference from other excited state dynamics. For our experiment, we chose acetonitrile, an ideal molecule with a relatively simple structure comprising just three hydrogen atoms, all bonded to the same carbon. As compared to previous work[25,26], this eliminates any ambiguity regarding the specific hydrogen atoms involved in H$_3^+$ formation, ensuring a singular pathway for this process. Additionally, considering acetonitrile's extensive use in industrial and chemical applications[33,34], gaining an in-depth understanding of its electronic and molecular reaction dynamics is critically important.

## Results and discussion

The schematic shown in Fig. 1 illustrates the dynamics induced in CH$_3$CN via the pump and probe pulses as well as the intermediate processes in our experiment. Using a femtosecond IR pump pulse, we excite CH$_3$CN to the dicationic state, where a neutral H$_2$ dissociates and roams in the vicinity of C$_2$NH$^{2+}$ due to the relatively flat potential energy surface. This forms our parent roaming channel, from which various fragmentation pathways emerge. Predominantly, we observe H$_2$ + H$^+$ + C$_2$N$^+$ from the parent roaming channel. A subset of the events from the parent channel undergoes proton transfer (PT), where the roaming H$_2$ captures a hydrogen ion from C$_2$NH$^{2+}$ to form H$_3^+$ + C$_2$N$^+$ [25,26,29]. Additionally, electron transfer (ET) from the roaming neutral H$_2$ to C$_2$NH$^{2+}$ may occur, resulting in H$_2^+$ + C$_2$NH$^+$ [29,31]. These competing pathways of PT and ET occur on the dicationic state with roaming H$_2$, independent of the probe pulse. On the other hand, interaction with the probe pulse can ionize the roaming H$_2$, leading to a triple-ion coincidence channel, H$^+$ + H$_2^+$ + C$_2$N$^+$, which disrupts both the PT and ET pathways. A detailed comparison of the relative yields of the parent roaming channel and the three product channels is discussed later in the article, focusing on the correlation and competition between the channels. All the channels of interest are determined in a kinematically complete manner as the momenta of all ionic fragments are measured in coincidence using the Cold Target Recoil Ion Momentum Spectrometer (COLTRIMS) technique[35,36]. A more in-depth discussion of our specific experimental parameters is given in the Methods section and a schematic our optical setup is shown in Supplementary Fig. 1 of the Supplementary Information (SI). We have additionally performed analogous measurements using deuterated acetonitrile, where all three hydrogen atoms are replaced by deuterium. This allows us to better resolve different masses of the detected charged particles. The deuterated or undeuterated measurements will be explicitly noted throughout the article, including a comparison of their temporal dynamics.

Our investigation into the roaming process starts with the experimental imaging of the underlying dynamics of roaming neutral D$_2$, the precursor to the formation of D$_3^+$. Here, we investigate the incomplete coincidence channel of D$^+$ and C$_2$N$^+$, using photoion-

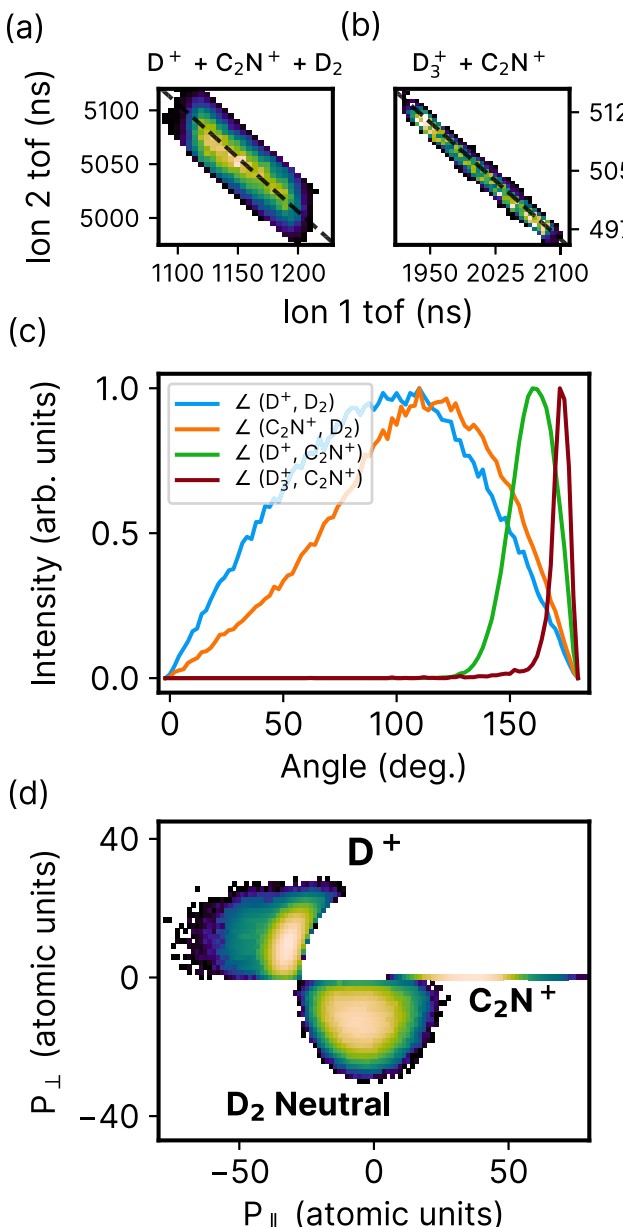

**Fig. 2 | Roaming coincidence channel $D^+ + C_2N^+ + D_2$ neutral.** Photoion-photoion coincidence lines for (**a**) $D^+ + C_2N^+ + D_2$ and (**b**) $D_3^+ + C_2N^+$ from acetonitrile, integrated over all pump-probe delays. x-axis shows the time-of-flight (tof) of the first ion, while the y-axis shows the tof of the second ion. Black dashed lines of slope -1 are overlaid on the PIPICO channels (**a**) and (**b**) to demonstrate the agreement between the expected and experimentally obtained slopes of these channels. **c** Angular distribution between the momentum vectors of $D^+$ and $D_2$ (blue line), $C_2N^+$ and $D_2$ (orange line), $D^+$ and $C_2N^+$ (green line), and $D_3^+$ and $C_2N^+$ (crimson line) within the first 200 fs time delay window. y-axis shows the intensity in arbitrary units for the different distributions. **d** Newton diagram for the channel $D^+ + D_2$ neutral $+ C_2N^+$ integrated over the first 200 fs of pump-probe delay. The momentum vector of $C_2N^+$ lies along the x-axis while those of $D^+$ and $D_2$ neutral are plotted in the top and bottom halves, respectively. The x- and y-axes show the components of the momentum vector (P) that are parallel and perpendicular to that of $C_2N^+$, respectively.

photoion coincidence (PIPICO) maps, shown in Fig. 2a. Such incomplete coincidence channels carry valuable information about the intrinsic dynamics in neutral fragmentations or events involving undetected fragments. Recent studies have started to explore such channels in small molecules, utilizing both CEI[29,30,37] and photoelectron

momentum distributions[38]. The incomplete PIPICO channel in Fig. 2a exhibits a distribution with slope of −1 and is broad due to the missing momentum carried away by the undetected fragment, $D_2$. Such broadening of the coincidence channel implies a case of deferred charge separation, where the neutral fragment is ejected first, followed by the remaining dicationic moiety breaking apart into two ions[39]. For comparison, the PIPICO map for the complete channel, $D_3^+$ and $C_2N^+$, is shown in Fig. 2b, where a very sharp distribution of slope −1 is observed due to momentum conservation.

An in-depth examination of the kinematic and time-resolved dynamics associated with this incomplete coincidence channel will enable us to disentangle the behavior of the roaming fragment. By leveraging the momentum conservation of the three fragments, the momentum vector and hence, the kinetic energy of $D_2$ can be reconstructed. The reconstructed kinetic energy, shown in Supplementary Fig. 2 of the SI, is less than 1 eV, which reinforces that the missing fragment $D_2$ in our incomplete channel is indeed neutral, and not an ion that was simply lost in the detection process. In Fig. 2c, we plot the angle between the momentum vectors of $D_2$ and $D^+$ (blue line), and $D_2$ and $C_2N^+$ (orange line). Overall, we see a large spread in the angular distributions between the neutral $D_2$ and the other two ions, implying a nearly random angle of emission for $D_2$, a key signature of roaming. This is because roaming fragments traverse relatively flat regions of the potential energy surface, which results in wide variations in their pathways, in contrast to direct dissociation, where the fragments follow a fixed MEP. By mapping the momentum vector of the neutral fragment, we thus have a more direct method for tracking roaming reactions.

In contrast to the roamer-ion angular distributions, in Fig. 2c, the ion-ion angular distribution for $D^+$ and $C_2N^+$ (green line) for this incomplete channel is much sharper and centered at ~160°, indicating strong Coulomb interaction resulting in a stronger momentum correlation. Additionally, for comparison, the angular distribution for $D_3^+$ and $C_2N^+$ (crimson line), which forms a complete two-body channel, is even narrower and centered at ~175°, indicating back-to-back emission, as would be expected from a two-body fragmentation process.

Beyond examining angular correlations between momentum vectors, we can also map the relative 3D momenta of the coincident fragments using Newton diagrams. In general, Newton diagrams are very useful tools for providing structural information about the reaction dynamics of a molecular system[40–42]. One of their unique features is the ability to distinguish different mechanisms of fragmentation: concerted, instantaneous fragmentation would appear as sharp, localized distributions in the Newton diagram[43], whereas sequential fragmentation would appear as a thin semicircle[42,44,45]. A typical Newton diagram for a three-body ion fragmentation channel in acetonitrile, $D^+ + D_2^+ + C_2N^+$, is shown in Supplementary Fig. 13 of the SI.

Figure 2d shows the Newton diagram for the channel $D^+ + D_2$ neutral $+ C_2N^+$. Here, the pump-probe delay is constrained to values less than 200 fs. The momentum vector of $C_2N^+$ is fixed along the x-axis, while those of $D^+$ and the reconstructed $D_2$ are plotted on the top and bottom halves, respectively. Due to momentum conservation amongst the three fragments, the Newton diagram lies on a plane and shows the relative momentum distributions of the two other fragments with respect to $C_2N^+$. Overall, several features of the $D_2$ neutral momentum distribution are markedly different from the all-ion channel ($D^+ + D_2^+ + C_2N^+$). Particularly, the distribution of the neutral $D_2$ in Fig. 2d shows a lack of clear angular dependence, in addition to a very broad, albeit lower momentum magnitude distribution. This implies that the neutral $D_2$ departs with a smaller fraction of the total momentum involved in this process, as would be expected from neutral fragmentation. This broad distribution of momentum magnitude of the neutral $D_2$ in conjunction with a lack of clear angular correlation is an unambiguous experimental signature for the roaming pathways. In contrast, the two ions in this incomplete channel, $D^+$ and

(a)

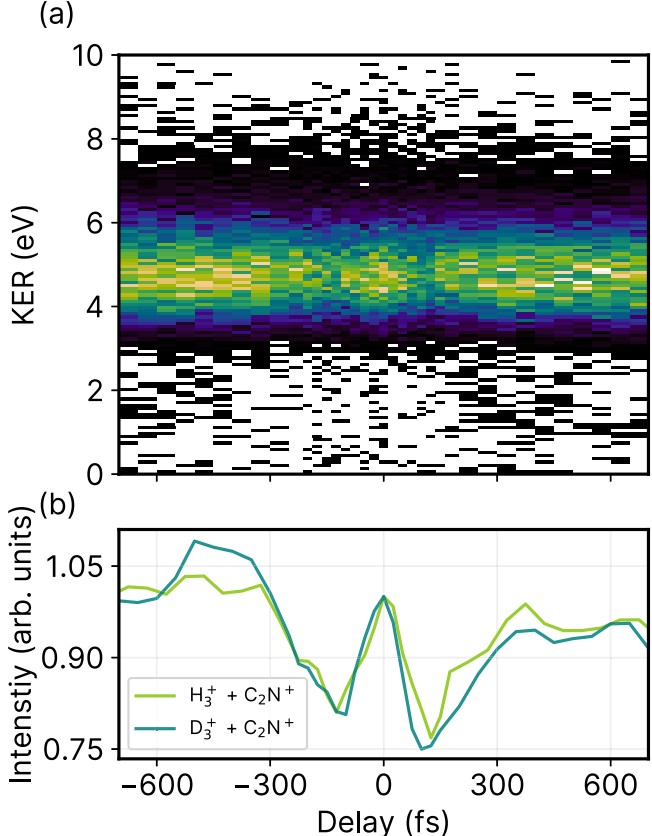

(b)

Intensity (arb. units)

Legend:
- H$_3^+$ + C$_2$N$^+$
- D$_3^+$ + C$_2$N$^+$

Delay (fs)

**Fig. 3 | Kinematically complete H$_3^+$ formation and timescale. a** Kinetic energy release of H$_3^+$ + C$_2$N$^+$ as a function of pump-probe delay. **b** Projection of the KER signal (between 3 eV and 7 eV) for H$_3^+$ + C$_2$N$^+$ (light green line) and D$_3^+$ + C$_2$N$^+$ (dark green line) on the pump-probe delay axis. The projected signal intensity in (**b**) is normalized to 1 at 0 fs pump-probe delay for both H$_3^+$ and D$_3^+$ channels and is presented in arbitrary units.

C$_2$N$^+$, share a stronger momentum correlation mediated via Coulomb interaction, as evidenced by a narrower angular spread along with a localized distribution of the magnitude of momentum. It is worth noting that in sequential processes, broad angular distributions can also arise from the rotation of the intermediate fragment. To eliminate any ambiguity, the Newton diagram in Fig. 2d is integrated over the first 200 fs pump-probe delay window, which is significantly shorter than the rotational timescale of the intermediate C$_2$ND$^{2+}$. As a result, the broad angular correlation that we observe here cannot be attributed to the rotation of C$_2$ND$^{2+}$ and is solely due to the roaming process. To further support this conclusion, Supplementary Fig. 14 in the SI shows the Newton plots of this channel for four time windows ranging from 0 - 800 fs, each integrated over 200 fs. In general, the distributions in the Newton plots are very similar for the entire delay range.

Thus far, we have only discussed the role of neutral D$_2$ in the formation of D$_3^+$. However, for a comprehensive exploration of the various pathways that can contribute to D$_3^+$ formation, we must also consider a roaming neutral D which can potentially abstract D$_2^+$. Interestingly, we do observe such an incomplete coincidence channel, which indicates the existence of a neutral D roaming pathway. Although the yield for this channel is relatively low due to the radical nature of a free H or D atom, making its formation less likely than H$_2$ or D$_2$, to the best of our knowledge, this is the first observation of such a pathway. A detailed analysis for this roaming D channel is shown in Supplementary Fig. 3 of the SI.

The technique of reconstruction of neutral fragment dynamics discussed here is very general and can also be applied for studying

other types of photochemical processes mediated by neutral fragmentation. For instance, in Supplementary Note IV of the SI, we study a different type of neutral fragmentation reaction in 2-propanol (CH$_3$CHOHCH$_3$). In this case, the fragmentation channel, CH$_3^+$ + H$_2$O neutral + C$_2$H$_3^+$, originates from two primary fragments, CH$_3^+$ and C$_2$H$_5$O$^+$, the latter of which subsequently dissociates into C$_2$H$_3^+$ and neutral H$_2$O. By comparing this case with neutral roaming, we observe significant differences in the PIPICO channel, angular distributions, and Newton diagram, which demonstrates that our approach provides distinct experimental signatures for various neutral fragmentation processes (see SI for more details).

After thoroughly investigating the dynamics of the D$_2$ neutral roamer, we now shift our focus to final photoproducts formed through such roaming reactions in acetonitrile, namely H$_3^+$ formation. Figure 3a shows the kinetic energy release (KER) map of the complete coincidence channel H$_3^+$ + C$_2$N$^+$ as a function of pump-probe delay. Here, we primarily observe a horizontal KER band centered around 5 eV along with a very weak time-dependent KER behavior. This distinctive constant KER behavior is due to a single pulse exciting the molecule directly into the dicationic state, which induces the formation of H$_3^+$ and C$_2$N$^+$. This is in contrast to the generally observed time-dependent decrease in kinetic energy which is due to the excitation of the system into an ionic state by the pump pulse and further ionization by the probe pulse.

Figure 3b shows the projection of the KER signal onto the pump-probe delay axis, highlighting the strong time dependence of the KER intensity. The observed features can be explained as follows. The enhancement of the total yield around time zero of the pump-probe delay is due to the overlap of the two pulses. On either side of time zero, we see an immediate decrease in the yield before the signal steadily increases again—this behavior is the result of the disruptive nature of the probe pulse that prevents H$_3^+$ formation by ionizing the neutral roamer H$_2$ before it can abstract a H$^+$. In other words, if the probe pulse arrives before H$^+$ abstraction, then it disrupts the roaming H$_2$, resulting in a decreased H$_3^+$ yield. Conversely, for longer time delays, the probe pulse arrives after H$^+$ abstraction. Thus, it is unable to disrupt H$_3^+$ formation, which results in an increase in its yield until it reaches saturation.

In Fig. 3b, we show the delay-dependent KER projections of the formation of both H$_3^+$ (light green line) and D$_3^+$ (dark green line). Overall, we observed no significant differences, within our experimental resolution, between the two isotopes of acetonitrile in the time-resolved measurements. This observation strongly indicates that the roaming process in this particular case relies mostly on electronic properties rather than on nuclear dynamics. This differs from other roaming reactions observed in deuterated methanol (CH$_3$OD)[25,46], where differences in timescale and KER are noted for H$_2$D$^+$ and H$_3^+$ formation. In CH$_3$OD, these product fragments are formed through two distinct pathways, potentially explaining the observed variations in timescale and KER. On the other hand, acetonitrile exhibits a singular pathway for H$_3^+$ or D$_3^+$ formation, providing a contrasting scenario.

We next fit the projections of the KER signal with a rising exponential of the form: $y = A(1 - e^{(t-t_0)/\tau}) + y_0$ where $A$ is the normalization constant, $t$ is the pump-probe delay, $t_0$ is the time offset, $\tau$ is the time constant, and $y_0$ is the intensity offset. $t_0$ and $\tau$ have physical significance since they reveal the timescales of the respective depletion and enhancement of the H$_3^+$ and D$_3^+$ formation. The values from the fitting function are given in Supplementary Table 1 in the SI while a comparison of the fits with the experimental data is given in Supplementary Fig. 5 of the SI. The average value of the time offset, $t_0$, is 125 fs, which is larger than the temporal resolution of the experiment (60 fs) and that of the time constant, $\tau$, is 105 fs. The fits indicate that H$_2$ roaming and H$_3^+$ formation occur on ultrafast timescales of approximately 200 fs, consistent with previously reported H$_3^+$ formation timescales in methanol, ethanol, acetone, and ethylene glycol[25,26,28].

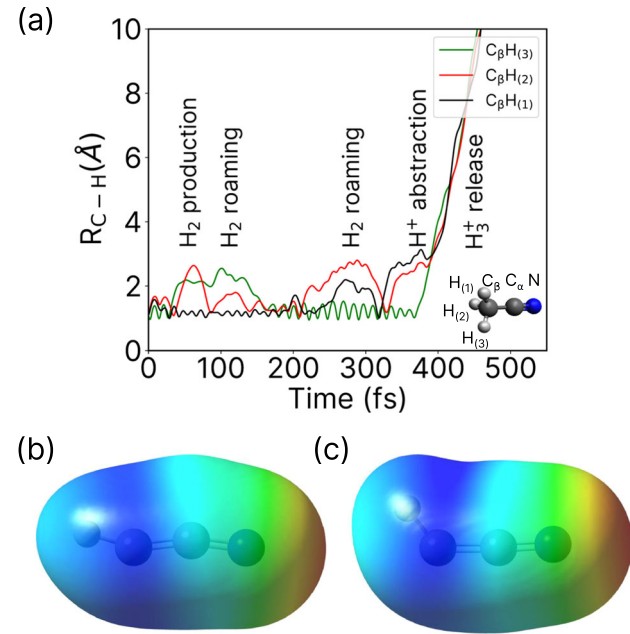

**Fig. 4 | Simulation of roaming $H_2$ and the corresponding potential energy surface. a** Time-dependent change of the three C–H bond distances for a trajectory showing $H_2$ roaming and $H^+$ abstraction leading to formation of $H_3^+$. **b**, **c** Electrostatic potential of $HCCN^{2+}$ mapped on the electronic density with isovalue 0.0004 a.u. Color code for the extreme values: red = 0.32 a.u. and dark blue = 0.48 a.u. The geometry used in (**b**) corresponds to the channel $H_2/HCCN^{2+}$, i.e., after releasing of neutral $H_2$, and in (**c**) to the weakly bonded $H_2...HCCN^{2+}$.

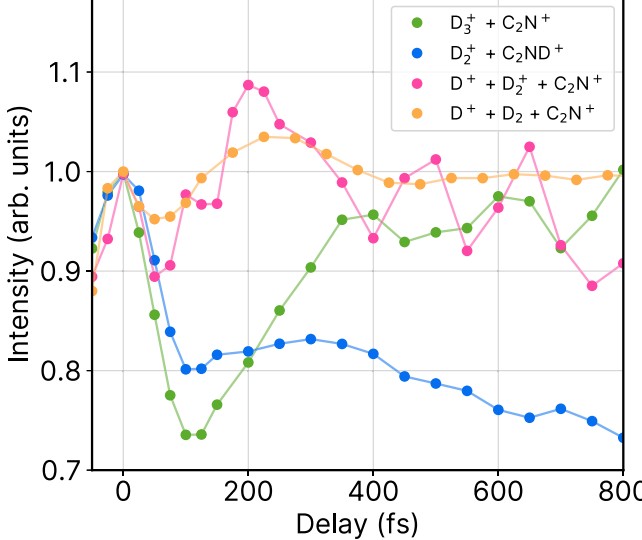

**Fig. 5 | Time-dependent signal intensities of competing channels.** Projection of the KER signals as a function of pump-probe delay for $D_3^+ + C_2N^+$ (green line) and $D^+ + D_2^+ + C_2N^+$ (magenta line) and $D_2^+ + C_2ND^+$ (blue line). Delay-dependent projection of the reconstructed roaming neutral $D_2$ (orange line) is also shown. The projections are normalized to 1 at 0 fs pump-probe delay and are presented in arbitrary units.

To better understand the mechanism of $H_2$ roaming and $H_3^+$ formation, we performed molecular dynamics simulations, described in detail in the Methods section. We find that the primary fragmentation channels producing cationic hydrogen species and their respective abundances are: $H^+/H_2C_2N^+ -79.6\%$, $H_2^+/HC_2N^+ -7.6\%$, and $H_3^+/C_2N^+ -0.8\%$. For the current work, we focus our analysis on the trajectories that produce $H_3^+$.

Figure 4 shows the time-evolution of the bond distances of the three H atoms relative to the molecular skeleton of CCN for a typical trajectory leading to $H_3^+$. We find that several processes occur within the first few hundred femtoseconds which impact $H_3^+$ formation. $H_2$ is produced in the first tens of fs and then roams up to ~150 fs. Subsequently, H exchange occurs, wherein the H atoms of the roaming $H_2$ molecule are bonded back to the carbon atom; this is immediately followed by a different pair of H atoms being ejected to form a new roaming $H_2$ molecule. After another ~200 fs, this $H_2$ molecule abstracts the last remaining H bonded to the C atom, thereby forming $H_3^+$ in coincidence with $C_2N^+$. Results for the other trajectories are given in Supplementary Figs. 10 and 11, and the corresponding variations of the nuclear kinetic energy with respect to time are given in Supplementary Fig. 12. In all cases, $H_3^+$ is formed via neutral $H_2$ roaming. From the limited number of simulated trajectories showing $H_3^+$ formation, we predict a timescale between 100 fs and 400 fs for this process, which agrees very well with the timescales extracted from the experimental measurements. The calculated nuclear kinetic energies in the final state are also in good agreement with the experimental KER (see Supplementary Fig. 12). Movies of four simulated molecular dynamics trajectories leading to $H_3^+$ formation, with snapshots at every 1 fs, are included as Supplementary Movie 1–4.

Further insight into $H_2$ roaming has been obtained via a careful exploration of the potential energy surface (PES). In particular, we computed the MEP that a doubly charged acetonitrile molecule would follow after vertical ionization, i.e., the maximum gradient on the PES, starting from the optimized geometry of the neutral system (shown in Supplementary Fig. 6). We clearly observe the release of neutral $H_2$, followed by the production of a weakly bonded $H_2...HCCN^{2+}$ complex. Furthermore, this shows that the neutral $H_2$ is polarized by the potential exerted by the dicationic fragment. We evaluated such potential by computing the electrostatic potential (ESP) of $HCCN^{2+}$ in two different configurations: using the geometry of the optimized structure from the moiety (Fig. 4b) and using the geometry of the last point given in the MEP exploration (Fig. 4c). The ESP is plotted in both cases by projecting the actual value on the electronic density isosurface, which provides a visualization of the potential that polarizes the neutral $H_2$. Overall, Fig. 4b, c reveal that the regions with the highest electrostatic potential (blue in color) are nearest to the C–H bond in $HCCN^{2+}$. It is most likely that the roaming $H_2$ remains weakly bound in this region. Upon fragmentation of $H^+$ and $CCN^+$, the $H_2$ is approximately at 90° with respect to the two ionic fragments, which qualitatively agrees with the angular correlations given in Fig. 2c. Supplementary Note VII in the SI provides additional detailed analysis on the roaming neutral $H_2$ trajectories obtained from molecular dynamics simulations.

By leveraging the power of coincident CEI, we can explore time-dependent correlations between the different fragmentation channels. In this case, each final product channel proceeds through the roaming channel, $C_2ND^{2+} + D_2$, which is created by the pump pulse. Figure 5 shows the yields of the predominantly observed roaming neutral $D_2$ in coincidence with the ionic parent fragments (orange line), along with $D_3^+$ formation through PT (green line), $D_2^+$ formation through ET (blue line), and three-body ionization through probe pulse interaction (magenta line). All channels show an initial peak at zero delay due to pump-probe overlap, followed by a decrease in signal, which varies depending on the channel.

The roaming neutral channel ($D_2 + D^+ + C_2N^+$) shows a subtle delay-dependent change in yield. It has slightly higher yield at short-time delays due to the increased ejection of the roaming neutral $D_2$, but this yield decreases slightly at longer delays as a result of branching into the final ionic product channels. Among them, the yield of the PT channel ($D_3^+ + C_2N^+$) initially shows a significant decrease before steadily increasing at later time delays. On the other hand, the three-body ionization channel ($D^+ + D_2^+ + C_2N^+$) shows the opposite trend,

with a sharp initial increase followed by a slow decay. The inverse correlation between these two channels is due to the direct competition between these fragmentation processes. Specifically, the formation of $D_2^+$, through the probe pulse ionization of neutral $D_2$, affects the yield of $D_3^+$.

Finally, the ET channel ($D_2^+ + C_2ND^+$) shows an enhanced signal intensity near zero delay, with a relatively low, constant yield up to 400 fs. This is likely due to the fact that the ET process is effectively suppressed in our experiment, either due to disruption by the probe pulse at shorter delays or large internuclear separations hindering ET at later delays[29].

In conclusion, we have developed a direct means to image neutral roaming reactions in small molecular systems. Using ultrafast IR pump-IR probe spectroscopy combined with coincident CEI, we have time-resolved the formation of $H_3^+$ from neutral $H_2$ roaming in acetonitrile, which was measured to occur within a few hundred femtoseconds. This technique enables us to directly track the invisible neutral roamer, the precursor to $H_3^+$ formation, from incomplete fragmentation channels. Additionally, with the aid of quantum chemistry calculations, we have fully simulated the possible roaming trajectories, allowing us to follow some of the unique intramolecular processes that occur in this type of dissociation. Our novel technique gives us a more straightforward means to observe neutral fragments which can allow us to gain a better understanding of the underlying molecular dynamics in roaming reactions. More generally, this provides a means to characterize neutral fragmentation processes, which are typically not possible with conventional spectroscopic techniques.

## Methods

A schematic of the experiment setup is given in Supplementary Fig. 1 of the SI, which has also been described in previous works[12,47,48]. We used a 5 kHz Ti:Sapphire laser producing 35 fs pulses with a central wavelength of 790 nm. The laser beam was split into two independent arms which were time-delayed with respect to one another. Each arm had an optical grating compressor to generate transform-limited pulses and had variable intensity controlled by a combination of $\lambda/2$ waveplate and polarizer. The pulse intensities were optimized to produce the highest contrast on the $H_3^+/D_3^+$ signal. For this experiment, the intensity of the pump and probe arms were respectively $3.2 \times 10^{14}$ W/cm$^2$ and $2.0 \times 10^{14}$ W/cm$^2$ with an uncertainty of ~10 %. The intensity was calibrated by measuring the relative ion yield ratios of doubly- to singly-ionized argon as a function of laser intensity. This was compared to a previously obtained reference yield ratio to determine the absolute peak intensity (see Supplementary Fig. 15 in the SI). The higher-intensity laser pulse served as the pump, and the weaker pulse as the probe for positive delays (vice-versa for negative delays). The two beams were directed into the interaction region where they were overlapped and back-focused to a spot size of about 10 $\mu$m, with a temporal resolution of about 60 fs. Both beams were linearly polarized in the direction of the time-of-flight axis of the spectrometer. A cold, molecular jet of acetonitrile, from a room-temperature bubbler, was produced by expansion through a 30 $\mu$m nozzle seeded with 1 bar of helium gas and propagated into the spectrometer perpendicular to the two laser pulses. Acetonitrile was ionized and resulting charged fragments were detected by a COLTRIMS[35,36]. Using a weak, homogeneous electric field, the ions were directed toward a position-sensitive detector, which is capable of measuring the three-dimensional momentum distributions of the charged particles. For CEI, we primarily focus on measuring the fragmented ions, and for the current experimental conditions, we have an ion momentum resolution of 0.1 a.u. By applying the coincidence technique, we can isolate specific mass channels of acetonitrile and gain the most relevant information about the fragmentation dynamics.

Ab initio molecular dynamics simulations were performed using the Atom Centered Density Matrix Propagation method[49–51] as implemented in the Gaussian16 program[52]. The electronic structure was computed by employing the density functional theory, in particular, the B3LYP functional[53,54] in combination with the 6−31++G(d,p) basis set[55,56]. To ensure the adiabaticity of the dynamics, we established a time step of $\Delta t = 0.1$ fs and a fictitious electron mass of $\mu = 0.1$ amu. Mimicking the experimental conditions, we considered a vertical double ionization of the acetonitrile molecule in a Franck−Condon manner, introducing a given amount of internal energy, $E_{exc} = 3$ eV, which was randomly redistributed over the nuclear degrees of freedom. This value was chosen to ensure that the energy available in the system agrees with the experimentally measured KER, and is discussed in more detail in Supplementary Note VIIIA of the SI. 500 trajectories were propagated up to 1 ps. We analyzed the statistical population of the different channels considering separated fragments with distances larger than 2.5 Å between atoms forming a moiety. Charges in the different atomic positions were computed from a Mulliken population analysis in the last step of each trajectory. We have successfully used this computational strategy in the past to infer the fragmentation dynamics in ionized molecules and clusters in the gas phase (see e.g. refs. [12,47,48,57–63]).

## Data availability

The raw, or suitably reconstructed, data are available from the corresponding authors upon request. Source data are provided with this paper.

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

## Acknowledgements

The experimental work was funded by the National Science Foundation under award No. 2306982. The theory was supported by the MICINN (Spanish Ministry of Science and Innovation) projects PID2019-105458RB-I00 and PID2022-138470NB-I00, funded by MCIN/AEI/10.13039/501100011033, the "Severo Ochoa" Programme for Centres of Excellence in R&D (CEX2020-001039-S) and the "María de Maeztu" Programme for Units of Excellence in R & D (CEX2018-000805-M). We acknowledge the generous allocation of computer time at the Centro de Computación Científica at the Universidad Autónoma de Madrid (CCC-UAM). We would like to acknowledge Noah Frese for his help in designing a figure.

## Author contributions

D.M. and A.C.L. conceived of and carried out the experiment in the ultrafast laser lab at the University of Connecticut. D.M., L.M.G., and A.C.L. analyzed the experimental data. S.D.-T. and F.M. performed the simulations and analyzed the theoretical results. D.M., A.C.L., S.D.-T., F.M., and N.B. interpreted the results and wrote the paper.

## Competing interests

The authors declare no competing interests.
