## [Peer Review File · Nature Communications]

Direct tracking of H₂ roaming reaction in real timeREVIEWER COMMENTS

Reviewer #1 (Remarks to the Author):

The manuscript reports new and interesting time-resolved measurements of the roaming H₂ dynamics in the acetonitrile dication. The roaming dynamics are initiated and probed by intense near IR laser pulses. In addition to measurements of the H₃⁺ + CCN⁺ product channel, the authors analyze two additional three-body channels, in which the missing mass and momentum are attributed to an "invisible" neutral H or H₂ fragment.

I have several comments that should be addressed before the work can be published:

1) In the abstract, the authors claim that the presented imaging approach is novel. Indeed, I am not aware of a similar implementation of imaging "invisible" neutral fragments in intense laser CEI of acetonitrile. However, the same approach was already described and implemented in very similar experiments with ultrafast EUV pulses on different organic systems (methanol, ethanol, 2-propanol etc...). Referring the readers to the existing literature will improve the manuscript and help convince the readers in the validity of the assignment of the missing mass and momenta to a neutral fragment.

2) The other interesting claim in the abstract is that the measured momentum of the neutral fragment in H₂ + CCN⁺ + H⁺ breakup allows direct reconstructing of the roaming H₂ momentum. Earlier works mentioned above have assigned the neutral H₂ (and H) fragments to a sequential fragmentation of one of the Coulomb explosion products. In methanol, this assignment was supported both by experimental and theoretical simulations. I would therefore ask the authors to discuss why the acetonitrile system behaves very differently. In particular as extensive simulations of roaming H₂ dynamics in methanol demonstrated that the neutral H₂ can not escape the pull of the dication. Do the simulated dynamics for acetonitrile indicate H₂ + H₂CCN²⁺ breakup of the dication?

3) On line 111, the text implies that H₂⁺ is formed by ionization of the roaming H₂. In contrast, in other organic dications methanol is formed also without a second probe laser. For example, in methanol, H₂⁺ is formed by electron transfer from the roaming H₂ to the dication and as shown in ref 27, a near IR probe pulse suppresses the roaming while enhancing 3-body fragmentation (as opposed to H₂⁺ formation). I did not notice a figure showing H₂⁺ enhancement by the probe pulse. Is there evidence for the roaming H₂ in the acetonitrile system to behave differently from methanol?

4) On line 103, the authors emphasize the lifting of ambiguity by using acetonitrile – one should maybe mention that several groups performed experiments on CH₃OD, lifting the ambiguity by deuteration and providing path selective timing information as well as evidence for a dynamical isotope effect in the KER following proton capture (see for example : <https://doi.org/10.1002/ntls.10022>

5) It will be very interesting to see if the KER resolution of the acetonitrile experiment allows observing the small shift in the KER that can be expected for D⁺ vs. H⁺ transfer.

6) On line 129 the authors suggest that the observed time dependence is in contrast to generally observed time resolved signal – I guess that they mean in system that exhibit sequential ionization by both pulses. While this is true, the pump probe signal in figure 2 is in very good agreement with the other works probing roaming H₂ in the methanol dication and in other similar systems.

7) On line 161 it is stated that there is no isotope effect on the roaming time – this is interesting. For positive times there seems to be a noticeable change on the scale that can be expected for the ultrafast dynamics (also confirmed by the fitting in the SI). The change is comparable to what one could expect from deuteration and is on the scale of the quoted 60fs resolution. Maybe the authors would like to revise their strong statement.

8) On line 176 it is stated that the roaming time is few hundred fs – while in the SI the fitting shows a ~ 100fs time scale that is actually in great agreement with simulated and experimental roaming H₂ times measurements in analogous systems.

9) The branching ratios quoted on line 183 show primarily 2-body breakup events. This is very different from other organic systems, which often exhibit 3-body breakup of the dication. Does this analysis include the ratio for 3-body breakup channels discussed in the manuscript – what is the branching ratio for those ?

10) On line 206 the text states that there's a limited number of H₃⁺ forming events – only 4 are shown in the SI. Is this out of the 500 simulated events ? Is the ratio comparable to the experimental 0.8 % ratio ? This is very different from roaming on the methanol dication that

exhibits ~ 10% branching ratios (shown both in experiment and theory – e.g. in ref 27). Is there a reason for the suppression of roaming H2 in acetonitrile?

11) On line 329, the authors argue that the angular distribution is broad due to the roaming and not due to rotational decoherence as the data is collected only for the 200fs time delays.

Analogous fragmentation in the CEI of several organic dications were attributed to sequential fragmentation, in which the decoherence can occur between the initial Coulomb explosion and the secondary fragmentation of one of the other fragments that can occur long after the time of the probe pulse. Here I again ask if this H2+ channel is observed also without the probe laser, can we see its time dependence?

12) In the last paragraph of results (~line 337) the authors describe a similar breakup in the isopropanol dication system. Indeed, Bittner et al, DOI: 10.1063/5.0098531, show 3-body analysis of 2-propanol CEI exhibiting both sequential and concerted 3-body breakup mechanisms that also help explain the suppression of the role of roaming in that system. A comparison of the strong-field CEI with the EUV data will be interesting.

13) Citation of references 16,17 have typos.

14) On line 67, its written that refs 25-27 are intense field studies that do not use the full power of CEI and rely heavily on theory. At least for ref 27 this is not quite correct, as it uses ultrafast EUV and although only one channel is highlighted - CEI is definitely used. Other CEI channels, including three-body breakup of a dication were discussed in other works (e.g. DOI:10.1126/sciadv.abq8084 , <https://doi.org/10.1021/acs.jpcllett.9b00576> , etc...)

15) In the introduction it should be noted that ref 28 is special as the CEI is implemented to study roaming dynamics on a neutral potential, rather on a dication state explored in other works. I suggest emphasizing this to point out that roaming H2 is ubiquitous not only in ionized systems.

16) Concerning the theory - Although DFT is often unsuitable for describing CEI dynamics. One should address in the text itself the addition of 3eV that was mentioned only in the methods section. In earlier simulation studies, roaming H2 dynamics was successfully reproduced in analogous systems (e.g. methanol), initialized from the Frank-Condon region also on the electronic ground state - The need for adding 3 eV makes the comparison less convincing of the correct theoretical interpretation. One can question if this is a difference between other organic dications and acetonitrile ? or is due to a problem with the level of theory (e.g. due to non single determinant nature, requiring use of CASPT2 or at least CASSCF for calculating the electronic potentials). It would be valuable also to show the result of the simulation without the additional 3eV, discussing the implication of the need to add 3eV.

Reviewer #2 (Remarks to the Author):

In this manuscript, the authors present a unique approach to directly observe the roaming phenomenon in molecular dissociation of acetonitrile (CH₃CN). They achieved this using femtosecond IR-IR pump probe spectroscopy and coincident momentum imaging technique. With their method the authors can directly track the neutral roaming fragments, H₂ in the present case, produced in the unimolecular decay of doubly ionized acetonitrile. Being neutral, these roamers are difficult to detect and study; however, in this study the momentum distributions of neutral H₂ were successfully reconstructed. To capture the dynamics of neutral H₂ is of significance importance, as it is a precursor to the formation of H₃⁺ during the dissociation of CH₃CN. Furthermore, the authors tested the effect of nuclear dynamics on this roaming process by using deuterated acetonitrile and concluded that this process is dependent on the electronic properties only. Their experimental results are supported by quantum chemistry based molecular dynamics simulations. There has been a considerable interest in exploring time-resolved roaming dynamics and in the formation dynamics of H₃⁺ in recent years. The phenomenon of "roaming" is particularly intriguing, as it involves a molecule that temporarily breaks into fragments which move independently before recombining thus deviating from conventional reaction pathways. Understanding this phenomenon holds the potential to provide valuable insights into the reaction mechanisms, energy distribution in specific dissociation pathways and the formation of products that may not be readily explained just by conventional theories. In this context, very recently Yang et al. explored the neutral H₂ formation in photoionized ethane using photoelectron coincidence spectroscopy [Nat Commun 14, 4951 (2023)]. A few years earlier, Endo et al. reported the

roaming dynamics of formaldehyde in real time using UV pump, NIR probe spectroscopy [Science 370,1072-1077(2020)]. Even on the surface of nanoparticles H₃⁺ formation was observed [Alghabra, M.S., Ali, R., Kim, V. et al. Anomalous formation of trihydrogen cations from water on nanoparticles. Nat Commun 12, 3839 (2021). <https://doi.org/10.1038/s41467-021-24175-9>]. Very recently, a new formation pathway for H₃⁺ starting from molecular dimers was reported [Nat. Chem. 15, 1224–1228 (2023) and Nat. Chem. 15, 1229–1235 (2023)]. In addition to these publications, there are many other reports on these topics that use extreme ultraviolet pulses. While this current work employs a well-established approach to experimentally investigate the dynamics of roaming molecular fragments in acetonitrile using pump-probe spectroscopy and COLTRIMS, the authors show that this time-resolved roaming dynamics can be directly captured by using Newton plots to tag the signal of neutral roamers. They explore the formation dynamics of H₃⁺ followed by the roaming dynamics of neutral H₂ and support their findings with theoretical simulations. They also demonstrate that this technique can be extended to obtain distinct experimental signatures for other neutral fragmentation processes by studying the neutral fragmentation of 2-propanol. Overall, the manuscript seems to be well-structured with all the experimental and simulation results appropriately interpreted. Considering the significance of this work, we think, this manuscript is suitable for publication in Nature Communications. However, there are a few major and minor presentation concerns that need to be addressed:

Major Concerns:

1. The authors have used linear polarization in this work. What is the significance of using this polarization and wavelength in this study? Did they try any other polarizations and wavelengths? Incorporating this discussion in the introduction section would be useful.

2. The main text requires a revision to explain the variation in kinetic energy signal intensity in Figure 2b. It is not clear, what do the authors mean by:

Line 145, "if probe pulse arrives early" and

Line 148 "for longer time delays"

It is important to ensure that the terminology is consistent throughout the text, especially with the explanation/caption of Figure 1. Furthermore, specifying the convention used to denote positive and negative delay would be highly valuable.

3. The details about the actual intensity in each arm are missing:

How was the intensity calibrated?

What other factors were considered to control the intensity in each arm?

Did the authors try to use equal intensity in each arm? If yes, why?

4. In Figure 3b, at around 370 fs the (green) proton suddenly shoots off, indicating a proton that is released just through internal relaxation. How efficient is the H₃⁺ formation compared to a H₂ + H⁺ pathway?

5. There is no discussion on the features of the reconstructed kinetic energy signal of the neutral roamer as a function of pump-probe delay in Figure 4c. It is important to include that.

Minor Concerns:

1. Including a time scale on Figure 1 and rephrasing the labels about how probe pulse affects H₃⁺ formation would be helpful in better comprehension.

2. The text in the last paragraph on page 3 (line 212 onwards) needs revision. It's not just the formation dynamics of D₃⁺ that was discussed in the preceding text.

3. It's worth discussing: why did they not include the data for the Newtons plots (Figure 4d) for in the manuscript?

4. Text in the paragraph starting at line 262 lacks the reference to the proper figure, which in this case should be Figure 4d.

5. In the context of highlighting the ability of newton plots to yield insight into dissociation dynamics we believe the following paper might be mentioned:

"Ultrafast Dissociation of Metastable CO₂⁺ in a Dimer."

<https://journals.aps.org/prl/abstract/10.1103/PhysRevLett.118.153001>

6. There is a typo in line 373, replace "power" with "intensity".

7. General comment on all the figures: Inconsistent font size on the axes labels. Also, surprisingly all 2d figures are missing color scales.

8. Legends on supplementary figures S6, S7 and S8 are missing.

9. Including a schematic diagram in the supplementary information would be beneficial for the readers.

10. Did the authors consider studying photoelectron momentum distributions to study this roaming phenomenon?

11. Finally, we suggest that this very recent article on neutral H₂ formation via intramolecular hydrogen migration in hydrocarbons in this journal should be mentioned:

"H₂ formation via non-Born-Oppenheimer hydrogen migration in photoionized ethane."
Nat Commun 14, 4951 (2023). <https://doi.org/10.1038/s41467-023-40628-9>

Finally, when reading up on the literature it occurred to us that the decay product C₂N⁺ might have technological significance as it is the monomer of a graphene-like 2-dimensional material [Tian, Z., López-Salas, N., Liu, C., Liu, T., Antonietti, M., C₂N: A Class of Covalent Frameworks with Unique Properties. Adv. Sci. 2020, 7, 2001767. <https://doi.org/10.1002/adv.202001767>]. The authors might wish to see if a link to 2-d materials can be made in order to broaden the impact of their work.

Reviewer #4 (Remarks to the Author):

The authors investigated the H₂-roaming induced H₃⁺ formation in CH₃CN molecule driven by femtosecond lasers. The results reported are so much overlapped with previous work on similar topic that I see no points attract me at all, see comment #2. Moreover, their interpretation is wrong. Friendly speaking, the article is premature. I do not think the paper deserves to be published.

1. The key point of this paper claimed by the authors is the visualization of the neutral fragments of H₂ in the H₃⁺ formation dynamics. The authors interpret the broadening in Fig.4b, angular distribution in Fig. 4d and the Newton plot in Fig. 4e result from the roaming of H₂. I don't agree! The roaming of H₂ is a sufficient condition that will lead to similar results, but the observation of such broadening is not a necessary condition to track back to H₂ roaming. The authors assign the detection of (D⁺, C₂N⁺) channel to the missed roaming D₂ from (D⁺, C₂N⁺, D₂) channel which can lead to H₃⁺ formation. There are several loopholes here.

1.1 Even if (D⁺, C₂N⁺) comes from (D⁺, C₂N⁺, D₂), it doesn't mean this is a roaming D₂ will lead to D₃⁺ formation finally. This is the reason why previous work (see comment #2) chose H₃⁺ or D₃⁺ as the observable.

1.2 Still, even if this (D⁺, C₂N⁺) comes from (D⁺, C₂N⁺, D₂), it can be a direct or sequential three-body dissociation channel. Molecules undergo severe structure deformation in strong laser fields, especially the intensity used here is saturated, see comment #3. The deformation of the molecular structure can also give rise to similar results for an incomplete fragmentation channel with neutrals. The incomplete fragmentation channel is much more complex than the complete one shown in Fig. S9.

1.3 (D⁺, C₂N⁺) can come from (D⁺, C₂N⁺, D, D). By judging from energy, the authors can exclude (D⁺, C₂N⁺, D⁺, D) and (D⁺, C₂N⁺, D⁺, D⁺), but not (D⁺, C₂N⁺, D, D). This channel will give rise to the same results in these plots.

The power of coincident measurements according to momentum conservation mostly works for complete fragmentation channels with all particle charged. For simple cases, e.g. diatomic or triatomic molecules, it also works if one knows the missing one has only one possibility. Specifically, here (D₂⁺, C₂N⁺) channel is easier to interpret than (D⁺, C₂N⁺), since only two choices of D⁺ or D for the former, but much more possibilities for the latter if one looks at incomplete coincident measurement. Regarding the above points, I don't think the interpretation is solid.

2. The present work overlaps so much with previous works, especially Sci. Rep. 7, 1 (2017) and Nat. Commun. 9, 1 (2018). Even if I assume the authors' interpretation was correct, still the novelty of this work does not deserve publication on Nature Communications. The research on

H3⁺ or D3⁺ formation dynamics seems to be a very hot topic recently. Nat. Commun. 12, 3839 (2021) is about its formation on the nanoparticle surface. Nat. Chem. 15, 1224–1228 (2023) and Nat. Chem. 15, 1229–1235 (2023) are about D3⁺ formation from D2-D2 dimer, accompanied with a News & Views Nat. Chem. 15, 1202–1203 (2023). The novelty of the above works are clear. Here, if the authors really shot the roaming movie of H2 that finally leads to H3⁺ formation, it will of course be interesting. But reading the manuscript, only a time-averaged Newton plot is there, with no time resolution and wrong interpretation.

There are four figures in the main text, the first three resemble those in previous work. The only difference is the authors use a different molecule here. The authors also know this, since in line 211, they said "After thoroughly investigating the formation dynamics of D⁺, we now shift our focus to the main goal of this work". Before they shifting their focus, they used already 3 figures! And their focus is only in figure 4. Why not move all these already reported results to the SI?

3. In Fig. 2 b, the authors showed the yield of H3⁺ and D3⁺ as a function of pump-probe delay. It is weird that at zero delay, where the pump and probe overlap so that the intensity is much stronger than positive or negative delays, the yield of H3⁺ or D3⁺ is in the same level as that in negative or positive delays. This contrasts with previous work, where the zero-delay always show the maximum yield due to high intensity. I think this is because the intensity for a single pulse is already saturated here. The authors only give a range rather than a specific value in the Methods.

4. Line 235, "Fig. 4 (c) shows the projection of the reconstructed kinetic energy of neutral D as a function of pump-probe delay." The authors can directly say "Fig. 4 (c) shows the yield of neutral D as a function of pump-probe delay." No matter which 2D plot one projects, that quantity is integrated and disappears, and one gets yield finally.

5. If the incomplete coincident channel really comes from or mostly comes from the roaming channels of (D⁺, C2N⁺, D2) that can lead to D3⁺ formation, Fig. 4c will show opposite tendency to Fig. 2b. But apparently not. This is a proof of self-contradictory on the authors' interpretation.

6. Line 323. "To eliminate any ambiguity, the Newton diagram in Fig. 4 (e) is integrated over the first 200 fs pump-probe delay window, which is significantly shorter than the rotational timescale of the intermediate C2ND2⁺." This argument is not true. The C2ND2⁺ can be excited to a very high rotational state when fragmentation.

Reviewer #1:

Comment:

The manuscript reports new and interesting time-resolved measurements of the roaming H₂ dynamics in the acetonitrile dication. The roaming dynamics are initiated and probed by intense near IR laser pulses. In addition to measurements of the H₃⁺ + CCN⁺ product channel, the authors analyze two additional three-body channels, in which the missing mass and momentum are attributed to an “invisible” neutral H or H₂ fragment.

I have several comments that should be addressed before the work can be published:

Response:

We would like to thank the reviewer for the careful reading of our manuscript and for the comments, which we address below:

Comment 1:

1) In the abstract, the authors claim that the presented imaging approach is novel. Indeed, I am not aware of a similar implementation of imaging “invisible” neutral fragments in intense laser CEI of acetonitrile. However, the same approach was already described and implemented in very similar experiments with ultrafast EUV pulses on different organic systems (methanol, ethanol, 2-propanol etc...) . Referring the readers to the existing literature will improve the manuscript and help convince the readers in the validity of the assignment of the missing mass and momenta to a neutral fragment.

Response:

We thank the reviewer for pointing out other bodies of work that leverage the power of studying incomplete channels with neutral fragmentation. We have added the following references in the paper (line 146-148), which use similar techniques, to strengthen our argument: K. Gope *et al.*, *Phys. Chem. Chem. Phys.* 25, 6979-6986 (2023), K. Gope *et al.*, *Sci. Adv.* 8, eabq8084 (2022), and D. M. Bittner *et. al.*, *J. Chem. Phys.* 157, 074309 (2022) .

Comment 2:

2) The other interesting claim in the abstract is that the measured momentum of the neutral fragment in H₂ + CCN⁺ + H⁺ breakup allows direct reconstructing of the roaming H₂ momentum. Earlier works mentioned above have assigned the neutral H₂ (and H) fragments to a sequential fragmentation of one of the Coulomb explosion products. In methanol, this assignment was supported both by experimental and theoretical simulations. I would therefore ask the authors to discuss why the acetonitrile system behaves very differently. In particular as extensive simulations of roaming H₂ dynamics in methanol demonstrated that the neutral H₂

can not escape the pull of the dication. Do the simulated dynamics for acetonitrile indicate $H_2 + H_2CCN^{2+}$ breakup of the dication?

Response:

The questions raised by the reviewer are quite interesting. In order to answer them, we have performed further analysis on the molecular dynamics simulations. Among the 500 trajectories, we have identified 38 which lead to the release of H_2 (7.6% of trajectories). In the supplementary information, we have included a discussion of this analysis (see new section VI of the SI). Briefly, in Fig. S6, we show the H_2 -HCCN distance as a function of time. The production of H_2 occurs in a time interval of 20 - 350 fs. In some of those cases, we clearly observe roaming trajectories. We have selected one of these trajectories and performed a detailed analysis (see Fig. S7 with snapshots and C-H distances). Here, we can see that the excitation energy is stored in C-H vibrational modes while the H_2 is roaming. At ~180 fs the vibrational energy is transferred to the H_2 , which is observed by the C-H vibrational amplitude decreasing while the kinetic energy of H_2 increases simultaneously. In other words, while H_2 is roaming, the C-H bond stores a significant amount of vibrational energy, and there is not enough translational kinetic energy to emit the H_2 . As soon as the energy is transferred from the vibrational mode of the C-H bond to the translation of H_2 , then emission occurs. It is impossible that an H_2^+ is the "roamer" since it would get expelled due to Coulomb repulsion. We can thus conclude that H_2 is produced after roaming. Fig. S8 shows the relevant stationary points in the potential energy surface where emission of H_2 is energetically accessible with a potential barrier. This is precisely the origin behind H_2 roaming before dissociation. In other words, the potential well keeps the H_2 weakly-bonded.

Finally, we believe that differences between methanol and acetonitrile can simply be explained by the difference in the molecular structure. While methanol is a rather floppy molecule with possible rotations along the C(sp³) and O(sp³) atoms, and with single (relatively weak) bonds, acetonitrile is quite a rigid molecule with linear skeleton along the heavy atoms CCN and with a very strong triple CN bond.

Comment 3:

3) On line 111, the text implies that H_2^+ is formed by ionization of the roaming H_2 . In contrast, in other organic dications methanol is formed also without a second probe laser. For example, in methanol, H_2^+ is formed by electron transfer from the roaming H_2 to the dication and as shown in ref 27, a near IR probe pulse suppresses the roaming while enhancing 3-body fragmentation (as opposed to H_2^+ formation). I did not notice a figure showing H_2^+ enhancement by the probe pulse. Is there evidence for the roaming H_2 in the acetonitrile system to behave differently from methanol?

Response:

We agree with the reviewer that $\text{H}_2^+/\text{D}_2^+$ can be formed via electron transfer from the roaming neutral H_2/D_2 . Below, we have listed the potential exit channels that can result from a neutral D_2 roaming around a C_2ND_2^+ fragment and Fig. 1 shows the time-dependent KER for each of these exit channels and Fig. 2 shows the normalized time-dependent yields of the KER :

Pump:

D_2 neutral + C_2ND_2^+

Exit channels when probe is absent:

Electron Transfer (ET): $\text{D}_2^+ + \text{C}_2\text{ND}^+$

Proton Transfer (PT): $\text{D}_3^+ + \text{C}_2\text{N}^+$

Exit channels when probe is present (interrupts both electron and proton transfer processes):

$\text{D}_2^+ + \text{C}_2\text{N}^+ + \text{D}^+$

Fig. 1: Pump-probe delay-dependent kinetic energy release (KER) for three complete channels: (a) $\text{D}_3^+ + \text{C}_2\text{N}^+$, (b) $\text{D}_2^+ + \text{C}_2\text{ND}^+$, and (c) $\text{D}^+ + \text{D}_2^+ + \text{C}_2\text{N}^+$

Fig. 2: Normalized time-dependent yields of the KER from $\text{D}_3^+ + \text{C}_2\text{N}^+$ (green), $\text{D}_2^+ + \text{C}_2\text{ND}^+$ (blue) and $\text{D}^+ + \text{D}_2^+ + \text{C}_2\text{N}^+$ (pink)

Fig. 2 above shows the normalized counts for the three exit channels. Comparing the time dependence of the energy-independent D_2^+ (3.1 - 5.8 eV) and D_3^+ (3.7 - 5.8 eV) yields provides insight into the competition between ET and PT. D_3^+ channel yield shows the distinctive signature of disruptive probing indicated by an initial decrease in signal intensity followed by an increase after a certain time-delay. This is a result of the probe pulse disrupting D_3^+ formation by ionizing the intermediate roaming neutral D_2 , leading to a 3-body coincidence channel of $D_2^+ + D^+ + C_2N^+$. We observe an increase in the 3-body channel yield between 50 and 350 fs which is correlated to the depletion in the yield of the D_3^+ channel.

The time-dependence of the D_2^+ yield is significantly different from the other channels. It remains fairly constant up to ~400 fs after which we observe a very slow decrease. To explain this behavior, we must consider the following competing pathways:

A) direct double ionization by a single pulse leading to D_2^+ and C_2ND^+ dissociation along the minimum energy path.

B) neutral D_2 roaming around C_2ND^{2+} , which eventually results in $D_2^+ + C_2ND^+$ via ET.

Pathway A's contribution, which is time-independent, sets the asymptotic limit for the D_2^+ yield. Conversely, at longer time-delays, the relatively large internuclear separation between fragments hinders pathway B. Additionally, at short time-delays, pathway B can be disrupted by probe pulse ionization of neutral D_2 . This complex interplay between pathways A and B, along with the probe's influence, modulates the signal intensity. Consequently, from the D_2^+ yield, we can conclude that the ET process is effectively suppressed, either due to the probe pulse or large internuclear separations.

Comment 4:

4) On line 103, the authors emphasize the lifting of ambiguity by using acetonitrile – one should maybe mention that several groups performed experiments on CH₃OD, lifting the ambiguity by deuteration and providing path selective timing information as well as evidence for a dynamical isotope effect in the KER following proton capture (see for example : <https://doi.org/10.1002/ntls.10022>

Response:

We thank the reviewer for highlighting prior works that disentangle the formation of H_3^+ or H_2D^+ using deuterated methanol. We have included the following citations (line 311-313) in our manuscript:

K. Gope, *et al.*, Nat Sci, **1**(2), e10022, 2021 [CH₃OD],

N. Ekanayake *et al.*, Sci. Rep., **7**, 4703 (2017).

Comment 5:

5) It will be very interesting to see if the KER resolution of the acetonitrile experiment allows observing the small shift in the KER that can be expected for D+ vs. H+ transfer.

Fig. 3: Normalized projections of KER for $H_3^+ + C_2N^+$ (dashed red) and $D_3^+ + C_2N^+$ (dashed green). The projection for each channel has been fitted to a normal distribution and the mean value is indicated by a dot-dashed line.

Fig. 3 above shows the projection of the KER distribution for H_3^+ (red) and D_3^+ (green) formation. The mean of the KER distribution of H_3^+ is at 4.81 eV compared to 4.71 eV for D_3^+ . Previous results [Gope et al. Natural Sciences, Ekanayake et al. Sci Rep 2017] report a difference of a few hundred meV in the median KER of H_3^+ and H_2D^+ formation (via ionization of CH_3OD) which is attributed to an isotope effect. However, for the case of CH_3OD , it must be noted that the H_2D^+ formation requires a longer pathway due to D⁺ abstraction from the hydroxyl. Additionally, it is energetically more favorable compared to H_3^+ formation since the H⁺ abstraction comes from the carboxyl group. In contrast, H_3^+ and D_3^+ formation in acetonitrile have identical reaction pathways and are energetically similar, which can potentially explain the nearly identical mean KERs for both these processes.

Additionally, the H_3^+ KER distribution is broader with a FWHM of 1.83 eV compared to 1.6 eV for D_3^+ . The higher KER for H_3^+ can be attributed to the relatively smaller interatomic distances between the H atoms in CH_3CN compared to the D atoms in CD_3CN . This smaller interatomic separation leads to a higher KER during Coulomb explosion.

Comment 6:

6) On line 129 the authors suggest that the observed time dependence is in contrast to generally observed time-resolved signal – I guess that they mean in system that exhibit sequential ionization by both pulses. While this is true, the pump-probe signal in figure 2 is in very good agreement with the other works probing roaming H2 in the methanol dication and in other similar systems.

Response:

The referee is right in that we were referring to the more typical time-resolved KER spectra, which show sequential ionization. We wanted to emphasize to general readers the distinctions observed in the 2D KER colormaps between conventional sequential ionization via pump and probe pulses, which yield time- and energy-dependent distributions, and roaming reactions which exhibit a time-dependent yet energy-independent distribution. While the yields in Fig. 3 (b) of the manuscript are in good qualitative agreement with previous results [Ekanayake *et al.*, Nat Commun., 9, 2018, Ekanayake *et al.*, Sci. Rep., 7, 2017], they primarily looked at time-resolved mass spectra as opposed to time-resolved kinematically complete KER spectra. For time-resolved mass spectra, it is not always clear which specific channel, or even charge states, the H_3^+ comes from. However, time-resolved KER spectra are intrinsically designated to a specific channel. Additionally, the KER doesn't change energetically as a function of pump-probe delay, which implies that all of the roaming dynamics are coming from the dicationic state.

Comment 7:

7) On line 161 it is stated that there is no isotope effect on the roaming time – this is interesting. For positive times there seems to be a noticeable change on the scale that can be expected for the ultrafast dynamics (also confirmed by the fitting in the SI). The change is comparable to what one could expect from deuteration and is on the scale of the quoted 60fs resolution. Maybe the authors would like to revise their strong statement.

Response:

We agree with the reviewer that the time constant, τ , for H_3^+ is approximately 50 fs shorter than that for D_3^+ at positive delays. Since the time scale difference falls within our experiment's temporal resolution and is negligible for negative delays, we have refrained from asserting a strong claim about seeing isotope effect in roaming. Nevertheless, we have now revised the sentence in the manuscript (lines 305-308) to state that we didn't observe a significant difference, within our experimental resolution. We would additionally note that observing an isotope shift is less likely for acetonitrile, compared to previous studies with alcohol molecules, since the roaming and abstraction occurs at the same molecular site.

Comment 8:

8) On line 176 it is stated that the roaming time is few hundred fs – while in the SI the fitting shows a ~ 100fs time scale that is actually in great agreement with simulated and experimental roaming H2 times measurements in analogous systems.

Response:

We agree with the reviewer that the observed timescales for H₂ and D₂ roaming in acetonitrile agree quite well with those reported in previous studies for methanol, ethanol, acetone, and ethylene glycol. We have stated this in lines 331-335 of the manuscript.

Comment 9:

9) The branching ratios quoted on line 183 show primarily 2-body breakup events. This is very different from other organic systems, which often exhibit 3-body breakup of the dication. Does this analysis include the ratio for 3-body breakup channels discussed in the manuscript – what is the branching ratio for those ?

Response:

The trajectories shown in the manuscript were performed with 3 eV of thermal excitation. Within this excitation energy the branching ratios after 1ps are: 1 fragment (no fragmentation but possible isomerization)= 12%; 2 fragments = 88%, i.e. we do not observe 3-body breakup channels. Again, the difference in fragmentation with respect to other organic molecules can be explained with the nature of the bonds in acetonitrile (see answer to comment #2).

Comment 10:

10) On line 206 the text states that there's a limited number of H3+ forming events – only 4 are shown in the SI. Is this out of the 500 simulated events ? Is the ratio comparable to the experimental 0.8 % ratio ? This is very different from roaming on the methanol dication that exhibits ~ 10% branching ratios (shown both in experiment and theory – e.g. in ref 27). Is there a reason for the suppression of roaming H2 in acetonitrile?

Response:

Relative yields are reported only amongst the three important two-body coincidence channels reported below:

Channel	Exp. yield	Relative exp. yield	Theory yield	Relative theory yield
H ⁺ + H ₂ C ₂ N ⁺	7,137,341	90.32	398	90.45
H ₂ ⁽⁺⁾ + HC ₂ N ⁺	736,730	9.32	38	8.64
H ₃ ⁺ + C ₂ N ⁺	28,421	0.36	4	0.91

As already mentioned, in the molecular dynamics simulations an excitation energy of 3 eV is introduced. We note that the good agreement with the experimental branching ratios indicates that the amount of excitation energy we chose is correct for this process.

Comment 11:

11) On line 329, the authors argue that the angular distribution is broad due to the roaming and not due to rotational decoherence as the data is collected only for the 200fs time delays. Analogous fragmentation in the CEI of several organic dications were attributed to sequential fragmentation, in which the decoherence can occur between the initial Coulomb explosion and the secondary fragmentation of one of the other fragments that can occur long after the time of the probe pulse. Here I again ask if this H₂⁺ channel is observed also without the probe laser, can we see its time dependence?

Response:

Fig. 4 shows the three Newton plots corresponding to three different types of fragmentation processes: a) three-body ionic fragmentation, b) two ions in coincidence with a neutral roaming fragment, and c) two ions in coincidence with a neutral, where the neutral is a secondary product formed from one of the two primary ionic fragments.

The distinction between various fragmentation processes becomes evident in the three-body ion momentum correlation plots (Newton diagrams). In Fig. 4 (a) and (b) above, we can clearly differentiate between ion momentum correlations in conventional fragmentation and neutral roaming processes. In the former, we observe sharp, localized ion momentum distributions, while in the latter, we see very broad correlations between the roaming neutral and the other two ions. These broad features are observed in both angular and momentum magnitude distributions of the neutral, which differ significantly from the signature of D_2^+ in Fig. 4 (a).

To further cement the connection between this broad, uncorrelated distribution and the roaming neutral fragment, we also examine another neutral fragmentation channel, as shown in Fig. 4 (c). This channel from 2-propanol involves an initial Coulomb explosion of two primary fragments, CH_3^+ and $\text{C}_2\text{H}_5\text{O}^+$, followed by a secondary delayed fragmentation of $\text{C}_2\text{H}_5\text{O}^+$ into C_2H_3^+ and neutral H_2O ; identical to the mechanism pointed out by the reviewer in their comment. The momentum distribution of neutral H_2O in Fig. 4 (c) is strongly directed towards C_2H_3^+ , confirming that H_2O results from a secondary decay of $\text{C}_2\text{H}_5\text{O}^+$. Moreover, this distribution is notably distinct from the momentum distribution of neutral D_2 in Fig. 4 (b), highlighting the versatility of the technique we used in capturing the intermediate dynamics of neutrals originating from various fragmentation processes.

With regards to H_2^+ formation without the probe pulse via electron transfer (ET), we have addressed this concern in the reviewer's third comment above. In our experiment, ET is effectively suppressed, either due to the disruptive probe or significant internuclear separations at longer time-delays that don't support ET. Furthermore, it is important to highlight that in generating the Newton diagram for the $\text{D}^+ + \text{C}_2\text{N}^+ + \text{D}_2$ neutral channel, we specifically consider events where D^+ and C_2N^+ are detected in coincidence with a missing D_2 mass. By applying this condition, we naturally exclude all events involving ET from D_2 , even if it is delayed, because in such cases, the resulting coincident end products would be D_2^+ and C_2ND^+ .

Comment 12:

12) In the last paragraph of results (~line 337) the authors describe a similar breakup in the isopropanol dication system. Indeed, Bittner et al, DOI: 10.1063/5.0098531, show 3-body analysis of 2-propanol CEI exhibiting both sequential and concerted 3-body breakup mechanisms that also help explain the suppression of the role of roaming in that system. A comparison of the strong-field CEI with the EUV data will be interesting.

Response:

We agree with the reviewer on the potential interest of a comparative study of fragmentation mechanisms using strong-field CEI and EUV. In this work, we specifically examined the $\text{CH}_3^+ + \text{C}_2\text{H}_3^+ + \text{H}_2\text{O}$ neutral channel, a sequential fragmentation process involving delayed secondary decay. With this, our goal was to highlight the versatility of our technique of neutral momentum reconstruction and its correlations with the coincident ions, providing insights into the intermediate dynamics of the neutral H_2/D_2 . However, in the recent work mentioned by the reviewer above, the authors report several other interesting sequential three-body breakup channels in 2-propanol using single-photon double ionization with EUV photons. These fragmentation channels are very promising for future exploration, where our analysis technique combined with strong-field CEI can potentially enhance our understanding of these processes.

Comment 13:

13) Citation of references 16,17 have typos.

Response:

We thank the reviewer for pointing these out. We have now corrected these citations in the manuscript.

Comment 14:

14) On line 67, its written that refs 25-27 are intense field studies that do not use the full power of CEI and rely heavily on theory. At least for ref 27 this is not quite correct, as it uses ultrafast EUV and although only one channel is highlighted - CEI is definitely used. Other CEI channels, including three-body breakup of a dication were discussed in other works (e.g. DOI:10.1126/sciadv.abq8084 , <https://doi.org/10.1021/acs.jpcclett.9b00576>, etc...)

Response:

We thank the referee for his comment. We have reworded this discussion in the introduction (lines 58-65) to properly address the different methods in which roaming can be initiated.

Comment 15:

15) In the introduction it should be noted that ref 28 is special as the CEI is implemented to study roaming dynamics on a neutral potential, rather on a dication state explored in other works. I suggest emphasizing this to point out that roaming H2 is ubiquitous not only in ionized systems.

Response:

We thank the reviewer for pointing out this important distinction. We have included a comment regarding this (line 66-68) in the manuscript.

Comment 16:

16) Concerning the theory - Although DFT is often unsuitable for describing CEI dynamics. One should address in the text itself the addition of 3eV that was mentioned only in the methods section. In earlier simulation studies, roaming H2 dynamics was successfully reproduced in analogous systems (e.g. methanol), initialized from the Frank-Condon region also on the electronic ground state - The need for adding 3 eV makes the comparison less convincing of the correct theoretical interpretation. One can question if this is a difference between other organic dications and acetonitrile ? or is due to a problem with the level of theory (e.g. due to non single determinant nature, requiring use of CASPT2 or at least CASSCF for calculating the electronic potentials). It would be valuable also to show the result of the simulation without the additional 3eV, discussing the implication of the need to add 3eV.

Response:

The question raised by the reviewer is quite pertinent. We have already mentioned the good agreement we have obtained with the experiment by introducing 3 eV of internal excitation energy. This shows that after double-ionization, some internal energy remains in the vibrational modes.

To confirm that one needs to run the simulations by introducing some amount of excitation energy, we have performed calculations for 500 trajectories in which the initial internal energy of the acetonitrile dication corresponds to the zero-point-energy of neutral acetonitrile, 1.23 eV. We are thus assuming ionization in the Frank-Condon region with a minimum amount of excitation energy. The results for the branching ratios are:

1 fragment (no fragmentation but possible isomerization) = 21.0%

$\text{H}^+/\text{H}_2\text{C}_2\text{N}^+ = 78.4\%$

$\text{H}_2/\text{HC}_2\text{N}^+ = 0.6\%$

These results do not explain the fragmentation observed in the experiment.

Again, the question concerning differences between acetonitrile with other organic molecules is found in the nature of the bonding. A more rigid structure with stronger bonds allows for the storage of larger amounts of excitation energy with a lower degree of fragmentation.

Reviewer #2:

Comment:

In this manuscript, the authors present a unique approach to directly observe the roaming phenomenon in molecular dissociation of acetonitrile (CH₃CN). They achieved this using femtosecond IR-IR pump probe spectroscopy and coincident momentum imaging technique. With their method the authors can directly track the neutral roaming fragments, H₂ in the present case, produced in the unimolecular decay of doubly ionized acetonitrile. Being neutral, these roamers are difficult to detect and study; however, in this study the momentum distributions of neutral H₂ were successfully reconstructed. To capture the dynamics of neutral H₂ is of significance importance, as it is a precursor to the formation of H₃⁺ during the dissociation of CH₃CN. Furthermore, the authors tested the effect of nuclear dynamics on this roaming process by using deuterated acetonitrile and concluded that this process is dependent on the electronic properties only. Their experimental results are supported by quantum chemistry based molecular dynamics simulations.

There has been a considerable interest in exploring time-resolved roaming dynamics and in the formation dynamics of H₃⁺ in recent years. The phenomenon of “roaming” is particularly intriguing, as it involves a molecule that temporarily breaks into fragments which move independently before recombining thus deviating from conventional reaction pathways. Understanding this phenomenon holds the potential to provide valuable insights into the reaction mechanisms, energy distribution in specific dissociation pathways and the formation of products that may not be readily explained just by conventional theories. In this context, very recently Yang et al. explored the neutral H₂ formation in photoionized ethane using photoelectron coincidence spectroscopy [Nat Commun 14, 4951 (2023)]. A few years earlier, Endo et al. reported the roaming dynamics of formaldehyde in real time using UV pump, NIR probe spectroscopy [Science 370,1072-1077(2020)]. Even on the surface of nanoparticles H₃⁺ formation was observed [Alghabra, M.S., Ali, R., Kim, V. et al. Anomalous formation of trihydrogen cations from water on nanoparticles. Nat Commun 12, 3839 (2021). <https://doi.org/10.1038/s41467-021-24175-9>]. Very recently, a new formation pathway for H₃⁺ starting from molecular dimers was reported [Nat. Chem. 15, 1224–1228 (2023) and Nat. Chem. 15, 1229–1235 (2023)]. In addition to these publications, there are many other reports on these topics that use extreme ultraviolet pulses.

While this current work employs a well-established approach to experimentally investigate the dynamics of roaming molecular fragments in acetonitrile using pump-probe spectroscopy and COLTRIMS, the authors show that this time-resolved roaming dynamics can be directly captured by using Newton plots to tag the signal of neutral roamers. They explore the formation dynamics of H₃⁺ followed by the roaming dynamics of neutral H₂ and support their findings with theoretical simulations. They also demonstrate that this technique can be extended to obtain distinct experimental signatures for other neutral fragmentation processes by studying the neutral fragmentation of 2-propanol. Overall, the manuscript seems to be well-structured

with all the experimental and simulation results appropriately interpreted. Considering the significance of this work, we think, this manuscript is suitable for publication in Nature Communications. However, there are a few major and minor presentation concerns that need to be addressed:

Response:

We would like to thank the reviewers for their careful reading of our manuscript and positive evaluation of our work. We have addressed their comments below:

Major Concerns:

Comment 1:

1. The authors have used linear polarization in this work. What is the significance of using this polarization and wavelength in this study? Did they try any other polarizations and wavelengths? Incorporating this discussion in the introduction section would be useful.

Response:

We thank the reviewers for this interesting point. For this particular experiment, we used a wavelength of 800 nm. The main reason for this is because the electric field of the laser couples strongly to the molecular system resulting in efficient ionization to charge states of +2 and even +3. For shorter wavelengths, the coupling is not as strong (Keldysh parameter) so ionization is not as efficient. Although one could choose a longer wavelength, this is technically complicated and would also result in a loss of pulse energy. Additionally, the polarization is not as important for this particular measurement since we are performing multiphoton ionization and measuring the ion momentum spectroscopically. In this case, the effect of polarization is minimal. In previous experiments using this technique, we have changed the orientation of the polarization, but did not observe any significant changes in the spectra.

We have added a brief discussion to the introduction (lines 58-68) where we describe the various ionization mechanisms at different wavelengths. We have avoided mentioning polarization because it does not necessarily fit into the context of the present paper.

Comment 2:

2. The main text requires a revision to explain the variation in kinetic energy signal intensity in Figure 2b. It is not clear, what do the authors mean by:

Line 145, “if probe pulse arrives early” and

Line 148 “for longer time delays”

It is important to ensure that the terminology is consistent throughout the text, especially with the explanation/caption of Figure 1. Furthermore, specifying the convention used to denote positive and negative delay would be highly valuable.

Response:

We agree with the referee that clarity is important. We have tried to make the revised manuscript more consistent in terminology. We have addressed the particular points listed above as well as made a general rewrite to check the terminology throughout the manuscript.

Comment 3:

3. The details about the actual intensity in each arm are missing:

How was the intensity calibrated?

What other factors were considered to control the intensity in each arm?

Did the authors try to use equal intensity in each arm? If yes, why?

Response:

The intensity of each arm was calibrated using a power meter. The pump and probe arm intensities were adjusted to optimize the yield for $\text{H}_3^+ + \text{C}_2\text{N}^+$ coincidence channel. Another crucial factor in determining the intensity was the ability to disrupt H_3^+ formation, leading to a depletion near time-zero for both positive and negative delays. In our experiment, the intensities for the pump and probe arms were $3.22 \times 10^{14} \text{ W/cm}^2$ and $2.03 \times 10^{14} \text{ W/cm}^2$, respectively. We have included these values in the Methods section of our manuscript.

Comment 4:

4. In Figure 3b, at around 370 fs the (green) proton suddenly shoots off, indicating a proton that is released just through internal relaxation. How efficient is the H_3^+ formation compared to a $\text{H}_2 + \text{H}^+$ pathway?

Response:

The trajectory in the former Fig. 3 (b) of the manuscript (now Fig. 4 (a) in the revised manuscript) shows H_3^+ formation at around 370 fs, indicated by all three C-H bond distances (green, red and black lines), increasing without bound.

Independently, we can assess the efficiency of H_3^+ formation by comparing its experimental yield to that of $\text{H}^+ + \text{H}_2$ neutral pathway. The normalized yield (total channel yield/total laser shots) for H_3^+ 0.0012 whereas $\text{H}^+ + \text{H}_2$ neutral has a yield of 0.0895. Consequently, H_3^+ formation is significantly less efficient compared to $\text{H}^+ + \text{H}_2$ neutral pathway.

Comment 5:

5. There is no discussion on the features of the reconstructed kinetic energy signal of the neutral roamer as a function of pump-probe delay in Figure 4c. It is important to include that.

Response:

We thank the referees for pointing this out. We realize the previous discussion was minimal. Due to the limited space in the manuscript, we have moved this data into the SI and included a longer discussion of how the time dynamics of the neutral roamer correlates to other fragmentation channels. We have additionally included the discussion and pertinent figure below.

The signal for neutral D_2 , which is reconstructed from the incomplete channel $D^+ + C_2N^+$ and shown in Fig. 5 (red), exhibits a slight increase in intensity at shorter time-delays. This implies an excess of neutral D_2 fragments – those that have neither formed D_3^+ yet nor been disrupted by the probe to form D_2^+ . Subsequently, this signal intensity decreases, indicating a decrease in population, potentially due to successful D_3^+ formation. The observed inverse correlation between the D_2 signal (red) and that of D_3^+ (green) strongly implies that the reconstructed neutral D_2 fragments are predominantly roaming in nature.

Fig. 5 shows the Projection of the KER signals as a function of pump-probe delay for $D_3^+ + C_2N^+$ and D_2 neutral. The projections are normalized to one at zero delay between the pump and probe pulses.

Minor Concerns:

Comment 1:

1. Including a time scale on Figure 1 and rephrasing the labels about how probe pulse affects H_3^+ formation would be helpful in better comprehension.

Response:

We have taken the reviewers' comment into consideration, and edited Fig. 1 in the manuscript to make the schematic more informative.

Comment 2:

2. The text in the last paragraph on page 3 (line 212 onwards) needs revision. It's not just the formation dynamics of D_3^+ that was discussed in the preceding text.

Response:

We thank the reviewer for pointing this out. We have significantly revised the manuscript to better present our results. The discussion mentioned in the comment (page 3, line 212 onwards) is now presented at the beginning of the manuscript.

Comment 3:

3. It's worth discussing: why did they not include the data for the Newtons plots (Figure 4d) for in the manuscript?

Response:

We are unclear about the reviewers' comment. In the former Fig. 4 (d) (now Fig. 2 (c) in the revised manuscript), we present angular correlation distributions among the momentum vectors of the three fragments. To create this plot, we consider only the $D^+ + C_2N^+ +$ neutral D_2 coincident events, as shown in the new Fig. 2 (a). Fig. 2 (d), the Newton diagram, shows correlations both between the angles and the magnitudes of the momentum vectors of the three fragments for all such events. Fig. 2 (c) is simply a histogram of the angles between the vectors, highlighting the distinction between ion-ion and ion-roaming neutral interactions. In other words, all of the data is included, we just represent the data in different ways to help the reader better see our results.

Comment 4:

4. Text in the paragraph starting at line 262 lacks the reference to the proper figure, which in this case should be Figure 4d.

Response:

We thank the reviewer for identifying this error. We have now included the correct reference to the figure in the manuscript.

Comment 5:

5. In the context of highlighting the ability of newton plots to yield insight into dissociation dynamics we believe the following paper might be mentioned: "Ultrafast Dissociation of

Metastable CO₂⁺ in a Dimer.”

<https://journals.aps.org/prl/abstract/10.1103/PhysRevLett.118.153001>

Response:

We thank the reviewers for bringing to our attention this work. We have now included this citation (#42) in our manuscript (line 204).

Comment 6:

6. There is a typo in line 373, replace “power” with “intensity”.

Response:

We have changed the wording in our manuscript.

Comment 7:

7. General comment on all the figures: Inconsistent font size on the axes labels. Also, surprisingly all 2d figures are missing color scales.

Response:

We have fixed all the font sizes on the axes label in our manuscript. In the 2D colormaps, color scales represent normalized counts for each discussed channel, but their standalone interpretation is limited. For meaningful comparison among two or more coincidence channels, as depicted in Fig. 5 of the revised manuscript or Fig. S9 in the SI, we normalize the channel yields to 1 at the time of zero pump-probe overlap.

Comment 8:

8. Legends on supplementary figures S6, S7 and S8 are missing.

Response:

We have now included the legends in the former Figs. S6, S7 and S8 (Figs. S10, S11 and S12 in the revised manuscript and SI).

Comment 9:

9. Including a schematic diagram in the supplementary information would be beneficial for the readers.

Response:

Fig. 1 of the manuscript serves this purpose by showing a schematic diagram for roaming neutral H₂ and its role in H₃⁺ formation. It is unclear what additional type of schematic the reviewers are asking for.

Comment 10:

10. Did the authors consider studying photoelectron momentum distributions to study this roaming phenomenon?

Response:

We appreciate the reviewers' question and agree with them that measuring the photoelectron momentum distribution can yield interesting information about the excited states involved in the different ionization processes.

Specifically, detection of photoelectrons in coincidence with ions using COLTRIMS is a highly differential measurement that allows for a method to probe the excited state molecular dynamics. That said, at 800 nm (1.5 eV), the momentum distributions can be very noisy due to strong field effects like tunnel ionization and above-threshold ionization. Although it's possible to get meaningful data despite these strong field effects (such as the paper mentioned in Comment 11), it becomes highly dependent on the size of the system and the levels of fragmentation. In other words, measuring photoelectron momentum distributions can work quite well for atoms and small molecules, but it can become tedious, if not impossible, for more complex systems and processes.

Comment 11:

11. Finally, we suggest that this very recent article on neutral H₂ formation via intramolecular hydrogen migration in hydrocarbons in this journal should be mentioned:

"H₂ formation via non-Born-Oppenheimer hydrogen migration in photoionized ethane."

Nat Commun 14, 4951 (2023). <https://doi.org/10.1038/s41467-023-40628-9>

Finally, when reading up on the literature it occurred to us that the decay product C₂N⁺ might have technological significance as it is the monomer of a graphene-like 2-dimensional material [Tian, Z., López-Salas, N., Liu, C., Liu, T., Antonietti, M., C₂N: A Class of Covalent Frameworks with Unique Properties. Adv. Sci. 2020, 7, 2001767. <https://doi.org/10.1002/advs.202001767>]. The authors might wish to see if a link to 2-d materials can be made in order to broaden the impact of their work.

Response:

We thank the reviewers for pointing out these references which certainly will help us broaden the impact of our work. We have now included the citation about H₂ formation (line 148) in our revised manuscript. We couldn't find a good way to add a discussion on 2D materials in the current manuscript. That said, we agree with the principle of broadening the impact of our work, so we have added a brief outlook in the Conclusion section.

Reviewer #4:

Comment:

The authors investigated the H₂-roaming induced H₃⁺ formation in CH₃CN molecule driven by femtosecond lasers. The results reported are so much overlapped with previous work on similar topic that I see no points attract me at all, see comment #2. Moreover, their interpretation is wrong. Friendly speaking, the article is premature. I do not think the paper deserves to be published.

Response:

We thank the reviewer for their work, however, we strongly disagree with their assessment. While earlier studies have focused on the final photoproducts of roaming, our work introduces a comprehensive approach. We present a complete molecular movie, starting with the formation and roaming of neutral fragments and concluding with the creation of H₃⁺. The direct tracking of neutral roamers is the missing link that offers an experimentally unambiguous signature for a process that is otherwise challenging to observe. With regard to the interpretation of our results, we have thoroughly addressed all of the reviewer's concerns with ample evidence supported by additional analysis results.

Comment 1:

1. The key point of this paper claimed by the authors is the visualization of the neutral fragments of H₂ in the H₃⁺ formation dynamics. The authors interpret the broadening in Fig.4b, angular distribution in Fig. 4d and the Newton plot in Fig. 4e result from the roaming of H₂. I don't agree! The roaming of H₂ is a sufficient condition that will lead to similar results, but the observation of such broadening is not a necessary condition to track back to H₂ roaming. The authors assign the detection of (D⁺, C₂N⁺) channel to the missed roaming D₂ from (D⁺, C₂N⁺, D₂) channel which can lead to H₃⁺ formation. There are several loopholes here.

Comment 1.1:

1.1 Even if (D⁺, C₂N⁺) comes from (D⁺, C₂N⁺, D₂), it doesn't mean this is a roaming D₂ will lead to D₃⁺ formation finally. This is the reason why previous work (see comment #2) chose H₃⁺ or D₃⁺ as the observable.

Response:

We agree that every roaming D₂ does not lead to D₃⁺; that is precisely the point of this work. Using coincident Coulomb explosion imaging, we can separate the process of roaming from roaming combined with D₃⁺ formation. The observable of D₃⁺ requires an additional step, D⁺ abstraction, which cannot be disentangled from the roaming step.

Comment 1.2:

1.2 Still, even if this (D+, C2N+) comes from (D+, C2N+, D2), it can be a direct or sequential three-body dissociation channel. Molecules undergo severe structure deformation in strong laser fields, especially the intensity used here is saturated, see comment #3. The deformation of the molecular structure can also give rise to similar results for an incomplete fragmentation channel with neutrals. The incomplete fragmentation channel is much more complex than the complete one shown in Fig. S9.

Response:

We disagree with the reviewer's point about laser intensity saturation. We have provided a more detailed response regarding this in our response to the reviewer's Comment #3.

We agree that three body dissociations can occur either directly or sequentially. Sequential processes, like deferred charge separation or secondary decay of primary fragments, amongst others, result in distinct coincidence channels with unique shapes and slopes (Laser Chem. 11, 259 (1991)). In our revised manuscript and SI, we discuss three distinct types of coincidence channels:

Fig. 2 (b): a complete two body channel in acetonitrile, $D_3^+ + C_2N^+$.

Fig. 2 (a): an incomplete two-ion coincidence, involving an undetected neutral fragment formed through deferred charge separation in acetonitrile, $D^+ + C_2N^+ + D_2$ neutral. In this channel, D_2 is ejected first, leaving behind C_2ND^{2+} , which subsequently undergoes further dissociation to give $D^+ + C_2N^+$.

Fig. S3 (a): an incomplete two-ion coincidence with an undetected neutral fragmentation formed via secondary decay of primary fragments in 2-propanol, $CH_3^+ + C_2H_3^+ + H_2O$ neutral.

In addition to the coincidence maps, we show the corresponding Newton diagrams associated with each of the three-body fragmentations discussed in the revised manuscript (Fig. 2 (d), Fig. S3 (c), and Fig. S13). They highlight the differences in ion and neutral momentum correlations based on whether the process is direct or sequential in nature. Specifically, since the (D^+ , C_2N^+ , D_2) channel is indeed sequential, we integrated the data from 0 to 200 fs to exclude any effects of rotation of the intermediate fragment as well as any potential structural deformation that can be induced in the molecule, both being effects that occur over longer timescales beyond 200 fs.

Comment 1.3:

1.3 (D^+ , C_2N^+) can come from (D^+ , C_2N^+ , D , D). By judging from energy, the authors can exclude (D^+ , C_2N^+ , D^+ , D) and (D^+ , C_2N^+ , D^+ , D^+), but not (D^+ , C_2N^+ , D , D). This channel will give rise to the same results in these plots.

The power of coincident measurements according to momentum conservation mostly works for complete fragmentation channels with all particles charged. For simple cases, e.g. diatomic or triatomic molecules, it also works if one knows the missing one has only one possibility. Specifically, here (D₂⁺, C₂N⁺) channel is easier to interpret than (D⁺, C₂N⁺), since only two choices of D⁺ or D for the former, but much more possibilities for the latter if one looks at incomplete coincident measurement. Regarding the above points, I don't think the interpretation is solid.

Response:

Fragmentation resulting in two free radicals (D⁺, C₂N⁺, D, D) is not a viable channel. Free radicals are one of the most chemically reactive substances in nature. A single free radical will immediately pair with whatever material is nearby, resulting in extremely short lifetimes. Two free radicals would immediately bond. For reference, please see *J. Chem. Phys.* **139**, 181103 (2013).

Comment 2:

2. The present work overlaps so much with previous works, especially *Sci. Rep.* 7, 1 (2017) and *Nat. Commun.* 9, 1 (2018). Even if I assume the authors' interpretation was correct, still the novelty of this work does not deserve publication on *Nature Communications*. The research on H₃⁺ or D₃⁺ formation dynamics seems to be a very hot topic recently. *Nat. Commun.* 12, 3839 (2021) is about its formation on the nanoparticle surface. *Nat. Chem.* 15, 1224–1228 (2023) and *Nat. Chem.* 15, 1229–1235 (2023) are about D₃⁺ formation from D₂-D₂ dimer, accompanied with a *News & Views Nat. Chem.* 15, 1202–1203 (2023). The novelty of the above works are clear. Here, if the authors really shot the roaming movie of H₂ that finally leads to H₃⁺ formation, it will of course be interesting. But reading the manuscript, only a time-averaged Newton plot is there, with no time resolution and wrong interpretation.

Response:

We do believe that our results are certainly complementary to the previous results on roaming shown in *Sci Rep* (2017) and *Nat. Comm.* (2018). That said, there are some dramatic differences in our work, in comparison to the previous works. First, the results that the referee mentions are simply time-resolved mass spectra, not time-resolved kinematically-complete, coincident measurements. In the former case, one cannot fully distinguish the fragmentation channel which forms the H₃⁺. This is clearly shown in our response to comment 3 below, which shows the yield of H₃⁺ is completely different from the yields shown in the new Fig. 3 (b). Additionally, the previous results used molecular systems where it is very difficult to disentangle where the roaming occurred within the molecule as well as the site of H⁺ abstraction. In our work, we have eliminated both ambiguities. Most importantly, we harnessed the full capabilities of the coincident Coulomb explosion imaging in order to directly reveal the intermediate step - random nature of the neutral roamer. This has never been shown before.

The referee's use of the term "time-averaged" when discussing the Newton diagrams in the paper is misleading. In general, when displaying pump-probe plots, one necessarily uses "time-averaged" results since a typical delay stage can move at steps far smaller than the temporal resolution. When performing high-order coincidence spectroscopy, one necessarily needs higher statistics. To our knowledge, Newton diagrams are almost always "time averaged". In this case, we show a time delay of 200 fs. Over this time range, we observed no distinguishing features in the Newton diagram and thus felt comfortable adding up the statistics. To state that these measurements have "no time resolution" misrepresents the actual data, which is clearly time-resolved. In fact, the time dynamics of the roaming neutral is even plotted in both the SI and in response to Comment 5.

There are four figures in the main text, the first three resemble those in previous work. The only difference is the authors use a different molecule here. The authors also know this, since in line 211, they said "After thoroughly investigating the formation dynamics of D⁺, we now shift our focus to the main goal of this work". Before they shifting their focus, they used already 3 figures! And their focus is only in figure 4. Why not move all these already reported results to the SI?

Response:

While we agree that the results in the former Fig. 4 are interesting, we disagree that the first three figures are unnecessary. The first figure is a schematic, which is commonly used to explain to the reader how the experiment was performed. The second figure showed the first time-resolved kinetic energy release distributions as a function of time delay. We note again that the results mentioned by the referee in earlier comments do not show H₃⁺ from a complete fragmentation channel. Those results are simply from a time-resolved mass spectrum. In that case, there are multiple pathways that can form an H₃⁺. The true power of coincidence spectroscopy lies in being able to extract kinematically complete information from specific fragmentation channels; which was not performed in the work mentioned by the referee. The third figure shows the simulations for our particular system, which, again, is necessary because these results are not simply an extension of the work that the referee has mentioned throughout their report. In fact, the interpretation used in previous works would not even fit here since all hydrogens are from a single site.

With that said, we have strongly restructured the paper in order to highlight the imaging of the roaming neutral. We hope that the referee appreciates the current way our data is presented.

Comment 3:

3. In Fig. 2 b, the authors showed the yield of H₃⁺ and D₃⁺ as a function of pump-probe delay. It is weird that at zero delay, where the pump and probe overlap so that the intensity is much

stronger than positive or negative delays, the yield of H₃⁺ or D₃⁺ is in the same level as that in negative or positive delays. This contrasts with previous work, where the zero-delay always show the maximum yield due to high intensity. I think this is because the intensity for a single pulse is already saturated here. The authors only give a range rather than a specific value in the Methods.

Response:

The reason for this discrepancy is not due to having too high laser intensity, resulting in a saturated signal. In general, it would be nearly impossible to even saturate a weak fragmentation channel like H₃⁺ since the target density is far too low. The reason for the discrepancy is that ours is a different measurement: we examine a kinematically complete fragmentation channel compared to the time-resolved mass spectra shown in Sci Rep (2017) and Nat. Comm. (2018). For comparison, we have plotted in Fig. 6 below, the H₃⁺ yield as a function of pump-probe delay for the same measurements that are now shown in Fig. 3 of the paper. Here, one can clearly see the increase in yield at time zero which is a similar effect as observed in Sci Rep (2017) and Nat. Comm. (2018). This shows that simply measuring time-resolved mass spectra is not sufficient enough to fully resolve a process like roaming. In particular, the differences in the yield between a time-resolved mass spectra and a kinematically complete fragmentation channel show that some of the H₃⁺ yield comes from other ionization/fragmentation processes in the former case.

Fig. 6: H₃⁺ yield as a function of pump-probe delay

Comment 4:

4. Line 235, “Fig. 4 (c) shows the projection of the reconstructed kinetic energy of neutral D as a function of pump-probe delay.” The authors can directly say “Fig. 4 (c) shows the yield of

neutral D as a function of pump-probe delay.” No matter which 2D plot one projects, that quantity is integrated and disappears, and one gets yield finally.

Response:

We think the wording suggested by the referee to be a little misleading. The term ‘yield’ would typically imply a measured quantity. For instance, the yield of $H_3^+ \rightarrow C_2N^+$ is given in the new Fig. 3 (b). The yield of neutral D_2 is not measured directly since our spectrometer relies on the measurement of charged particles. We believe it should be mentioned that this is a reconstructed quantity in order to make it abundantly clear to the reader that the neutral D_2 is not directly measured.

Comment 5:

5. If the incomplete coincident channel really comes from or mostly comes from the roaming channels of (D^+ , C_2N^+ , D_2) that can lead to D_3^+ formation, Fig. 4c will show opposite tendency to Fig. 2b. But apparently not. This is a proof of self-contradictory on the authors’ interpretation.

Response:

We thank the referee for pointing this out. Following their advice, in Fig. 7 below, we have plotted the time-dependent D_2 neutral (former Fig. 4c) along with the $D_3^+ + C_2N^+$ channel (former Fig. 2b) as well as the $D^+ + D_2^+ + C_2N^+$ channel. As can be seen, the roaming neutral indeed shows an inverse correlation to the $D_3^+ + C_2N^+$ channel. Therefore, our interpretation does not contradict either our data or the referee's perspective. Additionally, the $D_3^+ + C_2N^+$ channel is inversely correlated to the three-body $D^+ + D_2^+ + C_2N^+$ channel, which also explains that the role of the probe is to disrupt D_3^+ formation by ionizing neutral D_2 to form D_2^+ . We hope that this provides clarification for the referee. We have additionally included this figure in the SI along with a brief discussion since we believe this correlative behavior is relevant.

Fig. 7 shows the projection of the KER signals as a function of pump-probe delay for $D_3^+ + C_2N^+$, $D^+ + D_2^+ + C_2N^+$ and D_2 neutral. The projections are normalized to one at zero delay between the pump and probe pulses.

Comment 6:

6. Line 323. “To eliminate any ambiguity, the Newton diagram in Fig. 4 (e) is integrated over the first 200 fs pump-probe delay window, which is significantly shorter than the rotational timescale of the intermediate $C_2ND_2^+$.” This argument is not true. The $C_2ND_2^+$ can be excited to a very high rotational state when fragmentation.

Response:

We disagree with the referee’s point. Based on the published literature, molecular rotation for heavy systems like I_2 takes hundreds of picoseconds for high J states (see Nature **343**, 737 (1990)). Even for light diatomic molecules in extremely high J states (molecular superrotors), molecular rotation is on the picosecond timescale (see PRL **112**, 113004 (2014)). Therefore, $C_2ND_2^+$, an intermediate-sized fragment, should not undergo any rotation within the first 200 fs of photoexcitation.

REVIEWER COMMENTS

Reviewer #1 (Remarks to the Author):

In response to the review the authors made significant revisions to their manuscript and provided valuable additional information in their rebuttal letter. Unfortunately, in many of their responses the valuable information, including additional figures tables and discussion were provided only as a response to the review and not as actual revision in the manuscript.

One example is the additional information provided in response to comments 10 and 16 should be available to the readers to allow them as well as the reviewer an informed evaluation of the paper. Another example is that in their response letter the authors describe the competing ET mechanism in which the roaming can decay, as well as the surprising indication for ejection of the neutral H2 that was not observed by the earlier studies in other organic systems. However, these are important pathways that the reader must consider are not included in the simplified scheme in figure 1. Moreover, the figure suggests a single result of the probe pulse as a three cation Coulomb explosion, while the authors consider the $D^+ + C_2N^+ + \text{neutral } D_2$ as the result of the probe pulse.

Furthermore, the text implies that the $D_2 + D^+ + CND^+$ channel results primarily from the probe of the roaming H2. Nevertheless, the SI shows that it exhibits at most $\sim 5\%$ variation as a function of pump-probe delay. Thus, the selection of $< 200\text{fs}$ is not clear as most ($\sim 95\%$) of the contribution is independent of the time delay. No discussion is provided in the text to the fact that this channel can also arise from the product of the ET channel and not reflect the claimed "direct visualization" of the roaming H2.

In conclusion, in my opinion the discussion of the results must consider the other possible scenarios for the $D_2 + D^+ + CND^+$ observation. The evidence provided in their response that suggest that the neutral H2 can escape the dication are new and interesting, specially since it does not occur in the earlier studied systems exhibiting a roaming H2. Nevertheless, considering the competing processes I am not convinced that its observation can be described as a "direct visualization" of the roaming neutral D2. I suggest revising this ambitious statement repeated in the title and throughout the manuscript and to reconsider the implied "directness" of the relation of the measured velocities of the neutral D2 and the velocity of the roaming D2.

Reviewer #2 (Remarks to the Author):

The authors have diligently and in great detail addressed all the concerns raised by the referees during the initial review process. They have reorganized the manuscript to improve clarity and provided additional data in the revised manuscript and supplemental information to address the comments of all the reviewers.

However, while the rebuttal to comment #2 of Reviewer #4 about previous work was very convincing, the authors failed to make that point in the revised manuscript. We believe that by including this discussion in the introduction section, they can further convince the reader of the novelty of this work.

In response to our comments, the authors have incorporated largely adequate changes in the manuscript.

However, regarding our comment #3, the authors have not provided details on how they calibrate the laser intensity in the interaction region inside COLTRIMS. In their response they state: "The intensity of each arm was calibrated using a power meter." The power meter can just measure the power values not the intensity. One can calculate the peak intensity from the measured power and the other laser beam parameters, however, anyone being serious about intensity measurements will admit that this is a most unreliable method. The calculation assumes a perfect Gaussian beam profile, aberration-free focusing and exact knowledge of the pulse duration. We do not believe the authors have such an idealized situation. In our experience the calculation of the intensity from beam parameters overestimates the intensity typically by at least a factor of two when compared to in-situ calibration methods (<https://doi.org/10.1103/PhysRevA.70.023413>, <https://doi.org/10.1364/OE.19.009336>, <https://doi.org/10.1103/PhysRevA.87.053411>).

Furthermore, providing the value of peak intensity with a three digit-accuracy gives a false sense of precision. The laser is probably not stable over the course of the experiment on the percent level. Given that the authors probably did not calibrate the in-situ intensity, we expect them to at least define the calibration method ("using beam parameters") and add a reasonable margin of error to the quoted values.

To answer the second part of comment #3, although they mentioned the factors considered for adjusting the intensity in general, they didn't elaborate on the intensity control/optimization in each arm. This could be done with a neutral density filter, polarizer-half wave plate combination, or with a simple aperture.

In reference to our comment #9, we intended to request a schematic diagram of the experimental setup. We believe that a schematic of their specific experimental setup either in the supplemental material or even as a reference to a recent publication would be valuable for the reader.

The authors assert in their response to comment #11 that "We couldn't find a good way to add a discussion on 2D materials in the current manuscript. That said, we agree with the principle of broadening the impact of our work, so we have added a brief outlook in the Conclusion section". However, upon review, we did not identify changes in the conclusion section. It is possible that this statement was made in error or the authors may have forgotten to make amendments in the conclusion section.

When these last minor issues have been addressed, we can recommend publication without further reservations.

This is great work, and it represents an exciting step toward a better understanding the chemical process of roaming, which is fundamental for advancing theoretical and applied chemistry.

Reviewer #4 (Remarks to the Author):

It's nice that the authors supplement more data in the manuscript, which answers some of my doubts before. It's also good to see that the authors rephrased the manuscript, changing and reordering the figures and content accordingly. It's obvious that they realized the severe overlap with previous work, and now they emphasize more on their "new" results.

However, this really cannot be termed as new enough to me. I still insist that the result does not deserve Nature Communications. My suggestion is Communication Physics or scientific report.

Reviewer #1:

Comment:

In response to the review the authors made significant revisions to their manuscript and provided valuable additional information in their rebuttal letter.

Response:

We would like to thank the reviewer for their comments and suggestions. We have addressed all the concerns raised by them below.

Comment:

Unfortunately, in many of their responses the valuable information, including additional figures tables and discussion were provided only as a response to the review and not as actual revision in the manuscript. One example is the additional information provided in response to comments 10 and 16 should be available to the readers to allow them as well as the reviewer an informed evaluation of the paper.

Response:

We agree with the reviewer and have now included a detailed discussion in Supplementary Information Section VIII A. This discussion addresses the use of a 3 eV internal excitation energy, which results in a good match with our experimental findings (Comment 16). Additionally, we have included a table in this section that compares the relative channel yields obtained from both experimental and simulation results (Comment 10). The discussion is also briefly mentioned in the Methods section of the main text.

Comment:

Another example is that in their response letter the authors describe the competing ET mechanism in which the roaming can decay, as well as the surprising indication for ejection of the neutral H₂ that was not observed by the earlier studies in other organic systems. However, these are important pathways that the reader must consider are not included in the simplified scheme in figure 1.

Response:

We have now included ET as a potential decay pathway in our scheme in Fig. 1. We have additionally updated Fig. 5 in the manuscript and included a significantly longer discussion of the different channels that the neutral roamer can decay into. We hope this clarifies the work.

Comment:

Moreover, the figure suggests a single result of the probe pulse as a three cation Coulomb explosion, while the authors consider the D⁺ + C₂N⁺ + neutral D₂ as the result of the probe

pulse. Furthermore, the text implies that the $D_2 + D^+ + CND^+$ channel results primarily from the probe of the roaming H2. Nevertheless, the SI shows that it exhibits at most ~5% variation as a function of pump-probe delay. Thus, the selection of <200fs is not clear as most (~95%) of the contribution is independent of the time delay.

Response:

We would like to clarify that the formation of the D_2 neutral + $D^+ + C_2N^+$ channel is *not* a result of the probe pulse; the probe pulse disrupts this channel. The selection of a 200 fs time window for the Newton diagram was chosen to specifically exclude any rotational motion from the C_2ND^{2+} fragment creating the broad angular correlations that we observed.

In order to clarify that the neutral D_2 channel is not a result of the probe, we have included a few sentences just after the Introduction section of the manuscript. Additionally, we have explained the complex correlation between the four channels (including ET) by updating Fig. 5 and the corresponding text for the figure in the manuscript.

We would also like to note that the disruptive probe pulse was intentionally chosen to be weaker than the pump pulse. As a result, the variations of the neutral roamer as well as all other signals are relatively small (the largest being D_3^+ which has a 25% variation).

Comment:

No discussion is provided in the text to the fact that this channel can also arise from the product of the ET channel and not reflect the claimed “direct visualization” of the roaming H2.

Response:

We believe there may be a misunderstanding regarding the channel mentioned by the reviewer. We presume the reviewer refers to the D_2 neutral + $D^+ + C_2N^+$ channel, which they describe as possibly arising from the product of the electron transfer (ET) channel.

We must clarify that this D_2 neutral channel results exclusively from the pump pulse, as illustrated in Fig. 1 of our manuscript. Generally, electron transfer (ET) can lead to a $D_2^+ + C_2ND^+$ channel through the transfer of an electron from the roaming neutral D_2 to the intermediate fragment C_2ND^{2+} [K. Gope et al., *Sci. Adv.* 8, eabq8084 (2022)]. On the other hand, proton transfer (PT) represents a competing pathway, wherein the roaming neutral D_2 abstracts a D^+ from the intermediate dication to produce D_3^+ . These competing pathways were discussed in [K. Gope et al., *Sci. Adv.* 8, eabq8084 (2022), E. Wang et al., *J. Phys. Chem. Lett.* 14, 4372 (2023)]. The neutral roaming D_2 channel is the precursor to both ET and PT. In this work, we wish to directly track its temporal dynamics. Below is a flowchart (Fig. 1) and discussion about the different channels originating from roaming D_2 :

Fig. 1: Flowchart of parent and product channels showing the role of pump and probe pulses.

Parent channel:

D_2 neutral + DC_2N^{2+} : This channel is formed by pump pulse and decays to D_2 neutral + D^+ + C_2N^+ .

Product channels:

- (A) D_3^+ + C_2N^+ : Proton transfer channel formed from the parent ion. Lower yield at shorter delays since this channel gets disrupted by the probe, and higher yields at longer delays due to successful proton transfer.
- (B) D_2^+ + C_2ND^+ : Electron transfer channel formed from the parent ion. Low yield at shorter delays due to ionization of the parent channel by the probe interrupting electron transfer. At longer delays, the yield is even lower since ET is suppressed due to large

internuclear separations. A similar effect was observed in [K. Gope et al., *Sci. Adv.* 8, eabq8084 (2022)].

(C) $D^+ + D_2^+ + C_2N^+$: Triple-ion channel formed from the probe pulse ionization of the parent channel. Higher yield at shorter delays due to more available neutral D_2 that can be ionized by the probe, and lower yield at longer delays as a result of fewer available neutral D_2 , due to successful proton transfer.

In our work, we specifically investigate the parent neutral D_2 channel which has no other competing pathways. This approach allows us to bypass the complexities associated with competing processes like ET and PT.

We hope this clarifies this important point and helps the reviewer better appreciate how our current work fits into the broader research on this topic.

Comment:

In conclusion, in my opinion the discussion of the results must consider the other possible scenarios for the $D_2^+ D^+ + CND^+$ observation. The evidence provided in their response that suggest that the neutral H_2 can escape the dication are new and interesting, specially since it does not occur in the earlier studied systems exhibiting a roaming H_2 .

Response:

We thank the referee for recognizing the novelty of our results. We hope that the revised manuscript, especially the new Figs. 1 and 5 and the significant additions to the text, help to better elucidate the complex behavior that occurs during the roaming process.

Comment:

Nevertheless, considering the competing processes I am not convinced that its observation can be described as a “direct visualization” of the roaming neutral D_2 . I suggest revising this ambitious statement repeated in the title and throughout the manuscript and to reconsider the implied “directness” of the relation of the measured velocities of the neutral D_2 and the velocity of the roaming D_2 .

Response:

We hope that we have better clarified all of the competing processes above. It is important to note that the neutral D_2 channel should be considered the parent channel to electron or proton transfer. It is worth noting that not every roaming D_2 event leads to proton transfer or electron transfer, which is precisely the point of our work. Through the use of coincident Coulomb explosion imaging, we are able to distinguish between the roaming events and those resulting in formation of D_3^+ or D_2^+ . This effectively separates the process of pure roaming from secondary processes like electron or proton transfer. In the revised Fig. 5 and discussion, we

have presented the raw yield spectra for each product channel and have elucidated their complex correlated behavior.

Nevertheless, we have revised the use of “direct visualization” in the two instances it was used. We now use the term “tracking” to describe how we observe the nature of the roaming neutral.

Reviewers #2 and 3:

Comment:

The authors have diligently and in great detail addressed all the concerns raised by the referees during the initial review process. They have reorganized the manuscript to improve clarity and provided additional data in the revised manuscript and supplemental information to address the comments of all the reviewers.

However, while the rebuttal to comment #2 of Reviewer #4 about previous work was very convincing, the authors failed to make that point in the revised manuscript. We believe that by including this discussion in the introduction section, they can further convince the reader of the novelty of this work.

Response:

We agree that we can make a stronger statement in the paper as well. We have now included a better discussion of the novelty of our work in lines 73-78 of the manuscript.

Comment:

In response to our comments, the authors have incorporated largely adequate changes in the manuscript.

However, regarding our comment #3, the authors have not provided details on how they calibrate the laser intensity in the interaction region inside COLTRIMS. In their response they state: “The intensity of each arm was calibrated using a power meter.” The power meter can just measure the power values not the intensity. One can calculate the peak intensity from the measured power and the other laser beam parameters, however, anyone being serious about intensity measurements will admit that this is a most unreliable method. The calculation assumes a perfect Gaussian beam profile, aberration-free focusing and exact knowledge of the pulse duration. We do not believe the authors have such an idealized situation. In our experience the calculation of the intensity from beam parameters overestimates the intensity typically by at least a factor of two when compared to in-situ calibration methods (<https://doi.org/10.1103/PhysRevA.70.023413>, <https://doi.org/10.1364/OE.19.009336>, <https://doi.org/10.1103/PhysRevA.87.053411>). Furthermore, providing the value of peak intensity with a three digit-accuracy gives a false sense of precision. The laser is probably not stable over the course of the experiment on the percent level. Given that the authors probably did not calibrate the in-situ intensity, we expect them to at least define the calibration method (“using beam parameters”) and add a reasonable margin of error to the quoted values.

To answer the second part of comment #3, although they mentioned the factors considered for adjusting the intensity in general, they didn’t elaborate on the intensity control/optimization in

each arm. This could be done with a neutral density filter, polarizer-half wave plate combination, or with a simple aperture.

Response:

We apologize for the confusion. We thought the reviewers simply wanted the approximate pulse energy. To clarify here, we do not simply take the values for intensity from a power meter. For each experiment, we perform a calibration where we measure the ion yield ratio of singly- to doubly-ionized argon as a function of pulse energy. We then compare our results to a previously measured reference yield ratio in order to obtain the absolute peak intensity. Furthermore, while performing our measurement, we track the ion count rate over time to ensure that the pulse intensity remains constant. The approximate uncertainty is 10%. We have additionally added a section in the SI, which shows a schematic of our experimental setup and a description of the optical components that we use to control the intensity using a combination of half-wave plate and polarizer, and pulse length using grating compressors for each arm.

We have also added a paragraph to the Methods section to better explain how we measure the laser intensity.

Comment:

In reference to our comment #9, we intended to request a schematic diagram of the experimental setup. We believe that a schematic of their specific experimental setup either in the supplemental material or even as a reference to a recent publication would be valuable for the reader.

Response:

We have now included a schematic for our experimental setup in Section I of the SI. This schematic shows all of the important optical components of our IR-IR pump-probe setup.

Comment:

The authors assert in their response to comment #11 that “We couldn’t find a good way to add a discussion on 2D materials in the current manuscript. That said, we agree with the principle of broadening the impact of our work, so we have added a brief outlook in the Conclusion section”. However, upon review, we did not identify changes in the conclusion section. It is possible that this statement was made in error or the authors may have forgotten to make amendments in the conclusion section.

Response:

We apologize for the confusion. We wanted to keep the discussion in the Conclusion section on the broader impact of the technique. Namely, using our new method to distinguish roaming

by tracking the neutral fragment. In that effort, we had added a sentence at the end of our Conclusion section in the previous revised manuscript. However, we have now added a sentence (lines 113-116) along with references in the Introduction to mention how acetonitrile has broad industrial and chemical applications.

Comment:

When these last minor issues have been addressed, we can recommend publication without further reservations. This is great work, and it represents an exciting step toward a better understanding the chemical process of roaming, which is fundamental for advancing theoretical and applied chemistry.

Response:

We thank the reviewer for their appreciation of our results and for recommending it for publication in Nature Communication. We believe that this revised version of our manuscript includes all of the reviewer suggestions.

Reviewer #4:

Comment:

It's nice that the authors supplement more data in the manuscript, which answers some of my doubts before. It's also good to see that the authors rephrased the manuscript, changing and reordering the figures and content accordingly. It's obvious that they realized the severe overlap with previous work, and now they emphasize more on their "new" results.

However, this really cannot be termed as new enough to me. I still insist that the result does not deserve Nature Communications. My suggestion is Communication Physics or scientific report.

Response:

We thank the reviewer for praising our revised manuscript. We have toned down the wording while keeping true to our results, appreciated by the other referees.

REVIEWER COMMENTS

Reviewer #1 (Remarks to the Author):

The authors made significant improvements in the manuscript. Unfortunately, one point remains unclear to me. In figure 2d, the authors integrate over the first 200fs of pump-probe delay, this to avoid ambiguity due to rotation of the intermediate C_2ND^2+ . However, I have to apologize that I still can not see the relation the pump-probe delay time to the measured $D_2 + D^+ + C_2N^+$ correlations as in their response, and in the improved figure 1, the authors convincingly explain that the probe is only suppressing the roaming D_2 system by leading to triple ionization. Hence, even though the "probe" pulse arrives at 100 fs delay, the $D^+ + C_2N^+$ dissociation can still occur independently at a 500 fs delay or even longer times after the initial dissociation of the neutral D_2 . Indeed, the authors state with regard to figure 5 that the $(D_2 + D^+ + C_2N^+)$ shows only a subtle delay-dependent change in yield. Is there a significant change in the Newton diagram that forces selecting only events within 200fs time delays, and a probe delay dependent figure 2d? Regarding the discussion with reviewer #4, I tend to agree that the subject of roaming H_2 has been extensively studied in similar systems. I also argued that the momentum of the measured H_2 does not "directly visualize" the motion of the roaming H_2 . Nevertheless, as opposed to other systems reported in the literature, this system seems to exhibit ejection of a neutral H_2 – something that to the best of my knowledge was not previously observed by experiment or theory and therefore may deserve the attention of the growing community engaged in unraveling roaming dynamics and follows nature communications.

Reviewer #2 (Remarks to the Author):

After careful consideration of the revised manuscript and the author's thorough response to the concerns raised by the reviewers, we are pleased to accept this paper for publication in Nature Communications. The authors have shown a strong dedication to addressing the critiques and have significantly improved the clarity and completeness of the manuscript. However, we would like to reiterate the importance of including a proper reference when they describe how they calibrate intensity. Using the ratio of single to double ionization for intensity determination requires the use of a reference, because producing a calibrated intensity to ion yield curve is a non-trivial task. Hence, the authors should provide the reference that was used for their intensity calibration. While there may be a few minor issues that can be addressed through proofreading, the manuscript is in good shape and is ready for publication.

Reviewer #1:

Comment:

The authors made significant improvements in the manuscript.

Response:

We thank the reviewer for acknowledging the changes and improvements we have made to the manuscript as well as the figures. We hope that this better explains our work to the readers of Nature Communications.

Unfortunately, one point remains unclear to me.

In figure 2d, the authors integrate over the first 200fs of pump-probe delay, this to avoid ambiguity due to rotation of the intermediate C_2ND^2+ . However, I have to apologize that I still can not see the relation the pump-probe delay time to the measured $D_2 + D^+ + C_2N^+$ correlations as in their response, and in the improved figure 1, the authors convincingly explain that the probe is only suppressing the roaming D_2 system by leading to triple ionization. Hence, even though the “probe” pulse arrives at 100 fs delay, the $D^+ + C_2N^+$ dissociation can still occur independently at a 500 fs delay or even longer times after the initial dissociation of the neutral D_2 . Indeed, the authors state with regard to figure 5 that the $(D_2 + D^+ + C_2N^+)$ shows only a subtle delay-dependent change in yield. Is there a significant change in the Newton diagram that forces selecting only events within 200fs time delays, and a probe delay dependent figure 2d?

Response:

The reviewer is correct in saying that the $D^+ + C_2N^+$ fragmentation may occur well after the initial D_2 dissociation. However, the wide temporal range in the $D^+ + C_2N^+$ dissociation (anywhere between 0 fs to 1 ps) allows for the potential floppiness in the intermediate C_2ND^2+ structure, which can affect the angular distributions in our Newton plots. To minimize this influence, we focus on the first 200 fs, capturing events where C_2N^+ and D^+ fragment shortly after D_2 dissociation.

However, for completeness, below are the figures for Newton plots of the $D_2 + D^+ + C_2N^+$ channel integrated over different time-windows. We do not observe any significant difference in the angular distributions of the constituent fragments. As such, it is evident that C_2ND^2+ does not undergo any substantial rotation in this time-period. We note that this is not necessarily the case for some 3-body ion channels,

which can exhibit changes in both the magnitude and angle of their respective momentum vectors.

We have also included the Newton plots averaged over time-windows in the SI for completeness and added a sentence about them in the main manuscript.

Regarding the discussion with reviewer #4, I tend to agree that the subject of roaming H2 has been extensively studied in similar systems. I also argued that the momentum of the measured H2 does not “directly visualize” the motion of the roaming H2. Nevertheless, as opposed to other systems reported in the literature, this system seems to exhibit ejection of a neutral H2 – something that to the best of my knowledge was not previously observed by experiment or theory and therefore may deserve the attention of the growing community engaged in unraveling roaming dynamics and follows nature communications.

Response:

We thank the reviewer for recognizing the novelty of our work and supporting its publication in Nature Communications.

Reviewer #2:

Comment:

After careful consideration of the revised manuscript and the author's thorough response to the concerns raised by the reviewers, we are pleased to accept this paper for publication in Nature Communications. The authors have shown a strong dedication to addressing the critiques and have significantly improved the clarity and completeness of the manuscript.

Response:

We thank the reviewers for their support to publishing our work in Nature Communications as well as for acknowledging the substantial improvements we have made to the manuscript. We firmly believe these revisions have enhanced the overall quality of our submission.

However, we would like to reiterate the importance of including a proper reference when they describe how they calibrate intensity. Using the ratio of single to double ionization for intensity determination requires the use of a reference, because producing a calibrated intensity to ion yield curve is a non-trivial task. Hence, the authors should provide the reference that was used for their intensity calibration. While there may be a few minor issues that can be addressed through proofreading, the manuscript is in good shape and is ready for publication.

Response:

We understand the importance of having a proper calibration for the laser intensity. As stated previously, we measure the ratio of double to single ionization of a rare gas and compare it to our previous measurements, as well as published results. We typically perform this calibration a few times a year, for each set of experiments, to ensure our measured intensity is correct. Below is the yield ratio of doubly-charged to singly-charged Ar atoms as a function of laser intensity and the measurements agree with previous calibration measurements within a 5% error. For comparison, we have also included the measured Ar yield ratio given in C. Guo et al., Phys. Rev. A 58, R4271 (1998). As such, we are confident in the stated intensity which was given in the manuscript. We have included the laser intensity calibration data in the SI (Fig. S15) and hope that this satisfies the reviewers with regard to the quality of our measurements.

REVIEWERS' COMMENTS

Reviewer #1 (Remarks to the Author):

I think that the plots added to the SI confirm my suggestion that selecting a specific pump-probe delay does not help select specific "roaming times". In that respect the text of the manuscript may be a bit confusing to non-expert readers. I recommend that the authors reconsider the choice of words in the text. Nevertheless, by adding the figure to the SI the authors help readers avoid severe misinterpretation of the data and I can recommend the paper for publication. No further review is necessary.

Reviewer #1:***Comment:***

I think that the plots added to the SI confirm my suggestion that selecting a specific pump-probe delay does not help select specific "roaming times". In that respect the text of the manuscript may be a bit confusing to non-expert readers. I recommend that the authors reconsider the choice of words in the text. Nevertheless, by adding the figure to the SI the authors help readers avoid severe misinterpretation of the data and I can recommend the paper for publication. No further review is necessary.

Response:

We sincerely thank the reviewer for their careful and thoughtful review of our work. We would like to clarify that we never stated in our work that selecting the 200 fs time window for our Newton plot helps in choosing specific "roaming times." In fact, we have a sentence included in the manuscript stating: "In general, the distributions in the Newton plots are very similar for the entire delay range." Figure 14 in the SI supports this conclusion. We appreciate the reviewer's insights and are glad to see the paper recommended for publication.